

# The EAWS matrix, a look-up table to determine the regional avalanche danger level (Part B): Operational testing and use

Frank Techel[1], Karsten Müller[2], Christopher Marquardt[3], and Christoph Mitterer[3]

[1]WSL Institute for Snow and Avalanche Research SLF, Davos, Switzerland
[2]Norwegian Water Resources and Energy Directorate, Oslo, Norway
[3]Avalanche Warning Service Tirol, Innsbruck, Austria

**Correspondence:** Frank Techel (techel@slf.ch)

**Abstract.** To support public safety and risk management in snow-covered mountain regions, avalanche forecasts must deliver reliable and credible information on avalanche conditions. To promote greater consistency in avalanche danger level assessment across European avalanche warning services, a revised version of the EAWS Matrix – a structured look-up table that combines snowpack stability, the frequency of snowpack stability, and avalanche size to determine the regional danger level. Developed through expert elicitation, the Matrix includes cells with a single suggested danger level, while about half also display a second option, reflecting cases where a substantial minority of experts proposed a different level. We analyzed its operational use over the first three winters following implementation by 26 European avalanche warning services. Most Matrix cells were predominantly associated with a single danger level in operational use, suggesting potential for structural simplification. However, two cells – *poor–some–size 2* and *very poor–some–size 3* – acted as transition zones with substantial overlap between adjacent danger levels and divergent use across services. Assessments with finer granularity – such as sub-classes within the predefined Matrix categories – revealed meaningful tendencies within coarse classes and may help preserve critical nuances in expert judgment. Moreover, incorporating observed tendencies within these classes may enable more targeted guidance on when to assign the higher or lower of the two danger levels shown in the Matrix. Several Matrix cells remained rarely used, supporting the use of white shading to indicate uncertainty or implausibility in danger level assignment. While Matrix application was relatively consistent for dry-snow problems, marked inconsistencies emerged for wet- and gliding-snow problems, particularly in the classification of snowpack stability for the latter ones. These findings underscore the need for community-wide discussion and alignment of stability assessment practices and offer insights for refining both operational avalanche danger assessment and the Matrix itself. However, because neither the danger level nor its input factors can be independently measured, a true validation of Matrix performance remains out of reach. This study forms part of the iterative development process described in detail in the companion paper by Müller et al. (2025).

## 1 Introduction

Public avalanche forecasts play a key role in informing both authorities and recreational backcountry users about avalanche conditions at the regional scale. A primary objective of these forecasts is to communicate the severity of avalanche conditions clearly and efficiently (e.g., Engeset et al., 2018). To help users focus on the most important information first, avalanche



forecasts are structured according to the concept of an information pyramid, with the most relevant and actionable information placed at the top (EAWS, 2025d). The danger level – a single categorical value summarizing the hazard using a standardized five-level scale ranging from 1 (low) to 5 (very high) (EAWS, 2025b) – sits at the top of this pyramid. It offers an immediate, easily understood signal to communicate the severity of avalanche conditions and serves as the entry point to more detailed forecast content. Recreational backcountry users often rely on this information, particularly during the planning stage, to guide
their risk management strategies (e.g., Haegeli, 2010; Harvey et al., 2012; Schmudlach and Köhler, 2016).

  Consistent and accurate estimation of danger levels – both within and across regional or national warning services – is essential to their effectiveness and are prerequisites for providing value to end users (Murphy, 1993). Yet, forecasting avalanche danger at a regional scale is inherently complex. It requires synthesizing diverse data sources – including field observations and model predictions – that are often sparse, unevenly distributed in time and space, and available in both structured and
unstructured formats. This process culminates in an expert judgment of the danger level, integrating both the probability of avalanche occurrence and the expected size of potential avalanches in a region. As with most expert estimation tasks, differences in danger level assessments may occur even when the same information is available during the forecast process (e.g., Lazar et al., 2016; Techel et al., 2024). To improve the consistency and transparency of danger level assessments across Europe, the European Avalanche Warning Services (EAWS) introduced a revised framework. It includes updated definitions, a structured
operational workflow, and a new version of the EAWS-Matrix – a lookup table designed to support the determination of regional avalanche danger levels (EAWS, 2025a) (Figure 1).

  The revised EAWS-Matrix was developed through expert elicitation in a non-operational setting, using a survey-based approach, with the final version reached through expert consensus (Müller et al., 2025). Although a broad group of professional forecasters contributed to its development, the Matrix has not yet been systematically evaluated under real-world operational
conditions – where forecasts are issued under uncertainty and may have serious consequences if incorrect. Observing how the Matrix is used in operations can help demonstrate its practical value in supporting and harmonizing danger level assessments, while also revealing potential weaknesses in its structure. Ideally, such evaluation would consider both quality – whether the Matrix structure and input factors reflect real avalanche conditions – and consistency, in terms of how reliably it supports similar danger level assessments across services, which are key attributes of forecast goodness (Murphy, 1993). However,
since both the input factors and the resulting danger level are based on expert judgment and are not directly observable or measurable, such an evaluation is inherently difficult. Moreover, analyzing Matrix use in an operational context may reflect compliance with its design rather than an independent judgment of avalanche conditions – potentially masking design flaws.

  Given these limitations, we do not attempt to validate the Matrix in an absolute sense. Instead, we examine how it was used during daily operations to (i) identify differences in how avalanche danger is characterized using Matrix terminology across
warning services; (ii) analyze how the Matrix was applied across the range of issued danger levels; and (iii) explore differences between dry-snow and wet- or gliding-snow problems. These analyses provide insight into the practical implementation of the Matrix and highlight potential areas for refinement.

  The presented study is part of the iterative process described in the companion paper (Müller et al., 2025), which details the conceptual development of the revised Matrix and the accompanying workflow. Here, we focus on their operational implemen-



tation: how the Matrix was used in 26 warning services across multiple countries during the first three winters following its introduction, and how these real-world applications can inform ongoing improvements to the framework.

## 2   Determining the regional avalanche danger level – approach in Europe

While the methodological background and conceptual derivation of the Matrix are presented in detail in Müller et al. (2025), in the following we only briefly review the key principles of regional avalanche danger assessment in Europe.

To support more consistent and transparent assessments of regional avalanche danger across warning services, the European Avalanche Warning Services (EAWS) introduced a revised conceptual framework for their operations. It includes updated definitions of the factors (snowpack stability, frequency of snowpack stability, and avalanche size) determining avalanche danger, a structured operational workflow, and the EAWS-Matrix (EAWS, 2025a). While conceptually aligned with the Conceptual Model of Avalanche Hazard (CMAH, Statham et al., 2018), the framework is tailored to the regional scale, where avalanche

danger must be evaluated across entire warning regions encompassing diverse terrain types, slope aspects, and elevation bands – often ranging from below treeline to high alpine – and culminating in the determination and communication of a danger level.

The workflow begins by identifying all relevant avalanche problems (EAWS, 2025c). For each problem, forecasters assess both its presence (including affected slope aspects and elevation bands) and its contribution to the overall danger. This includes estimating the probability of avalanche occurrence and avalanche size. The avalanche occurrence probability is further

decomposed into two components (EAWS, 2025a):

- **Snowpack stability** describes the local propensity of the snowpack to avalanche (Reuter and Schweizer, 2018) and is categorized into four classes: *very poor*, *poor*, *fair*, and *good* (Table A1). These classes correspond to typical triggering mechanisms. For instance, avalanches releasing due to natural causes – such as loading from new snow or weakening from rain or melt water – are linked to *very poor* stability, while *poor* stability is commonly associated with human

triggering.

- **Frequency of snowpack stability (classes)** refers to the proportion of avalanche terrain within a region where a given stability class occurs. Frequency is categorized into four classes: *many*, *some*, *a few*, and *none or nearly none* (Table A2). The last class indicates that the stability class is either absent or so rare that it is not considered relevant for the avalanche danger level assessment.

In practice, estimating the full spatial distribution of snowpack stability across a region – where conditions vary by elevation, aspect, and slope – is not feasible (Techel et al., 2020). In fact, forecasters must synthesize limited information to estimate the prevalence of the lowest stability class(es), which forms the basis for judging the probability of avalanche occurrence. Combined with the largest avalanche size to be reckoned with, categorized into five classes (Table A3), these factors inform the danger level. Thus, the three inputs – stability, frequency, and avalanche size – constitute the core of the EAWS-Matrix.

The EAWS-Matrix is a look-up table providing visual guidance for assigning the danger level based on these assessments (Figure 1). It consists of three panels – one each for the lowest relevant stability class (*very poor*, *poor*, and *fair*) – with each



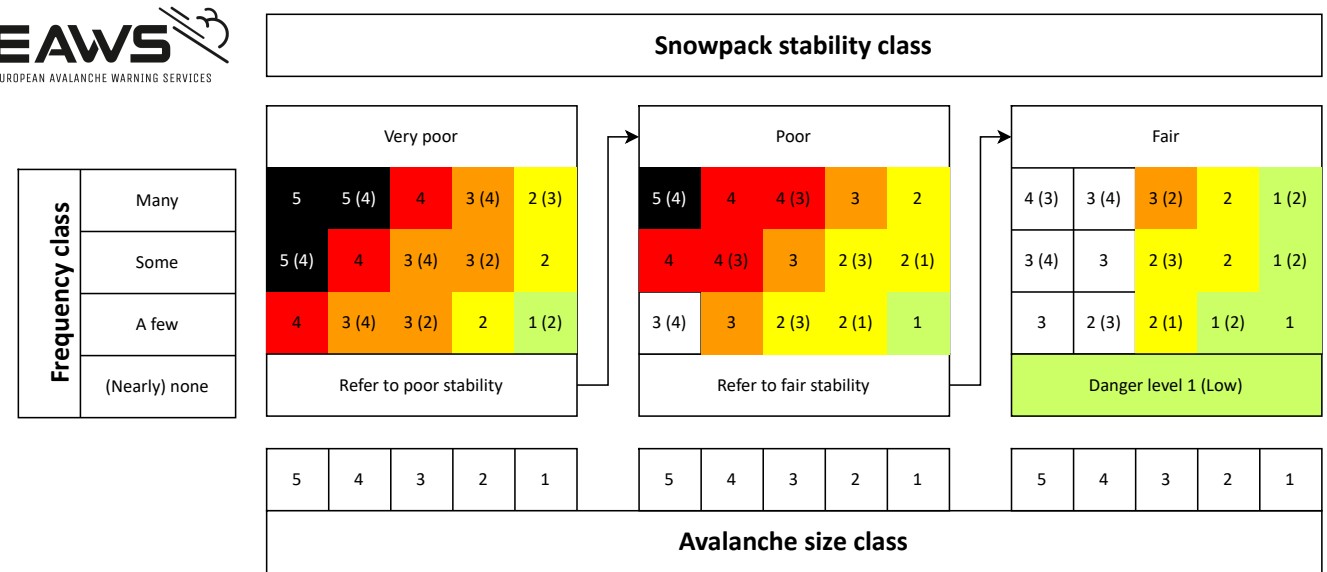

**Figure 1.** EAWS-Matrix, as accepted by the EAWS General Assembly in 2022 (taken from EAWS, 2025a). The integer values shown in the Matrix cells refer to the danger levels. We refer to the first danger level shown in the Matrix as $D^1$, and to the danger level shown in brackets as $D^2$. For a detailed explanation refer to the text.

showing combinations of frequency (y-axis) and avalanche size (x-axis). Each cell contains one or two danger levels: the primary value ($D^1$), shown first and indicated by the cell's color, represents the majority expert view and is referred to as the Matrix-suggested danger level. A secondary value ($D^2$), shown in brackets, is included when at least 30% of experts selected

a different level (EAWS, 2025b; Müller et al., 2025).

To determine the danger level, forecasters begin with the lowest assessed stability class and evaluate whether its frequency is relevant (i.e., not *none or nearly none*). If not, they proceed to the next stability class, which is congruent with an increase in triggering level needed to release an avalanche, following the arrows between panels (Figure 1), and so on. If stability is assessed as *good* throughout the region, the danger level is set to 1 (low) by default. Forecasters locate the Matrix cell that

best represents the conditions for each avalanche problem. Avalanche size is assessed independently and reflects the largest avalanche that could reasonably be reckoned with under the expected conditions.

The final communicated danger level for a warning region is the highest level assigned across all relevant avalanche problems. When different problems occur in a way that the combined frequency of potential triggering spots increases in the same aspects and elevations, their combined effect may result in a higher overall danger level than any single problem would suggest

in isolation (Müller et al., 2025).





## 3  Data

### 3.1  Data overview

In total, 26 avalanche warning services recorded their factor choices (snowpack stability, frequency of the lowest snowpack stability class, expected avalanche size) during operational forecasting, covering one or more winter seasons or extended parts
thereof. Table 1 provides an overview, including the three-letter abbreviations used when referring to specific warning services.

### 3.2  Varying forecasting practices and data harmonization

Forecasting practices varied both across and within services over the three-year study period, requiring careful standardization to enable meaningful comparisons. These differences concerned not only the format of the data provided for this analysis, but also how the EAWS-Matrix and the factor assessments were applied in practice. While we do not have detailed knowledge of
service-specific procedures, the following examples illustrate key variations – some of which may reflect differences in data availability rather than differences in actual workflows – and how we addressed them.

**Avalanche problem-specific assessments:** Most services assessed each avalanche problem individually and assigned it a corresponding danger level. However, some issued a single overarching danger level despite assessing factors per problem (e.g., BAY in first year), grouped problems into broader categories (e.g., SWI separated dry-snow from wet- or gliding-snow
problems), or assessed the factors without assigning them to a specific problem (e.g., SCO). In certain services (e.g., SWE), no factor assessments were made when no avalanche problem was identified – a situation commonly observed for danger level 1 (low). Others documented such situations using the label *no distinct avalanche problem* (e.g., SVK, SWI).

**Matrix integration in workflow:** Most services had integrated the EAWS-Matrix into their operational workflows – either using it to directly link factor assessments to a suggested danger level or as a reference framework. A notable exception was
Switzerland (SWI), where forecasters assessed factors and danger levels independently, with the latter determined during group discussions rather than derived from the Matrix (Winkler et al., 2024).

**Assessment of Matrix input factors:** The majority of services used the EAWS-defined ordinal classes for estimating the three Matrix input factors (EAWS, 2025a) (see Table 2), though some used North American terminology (as defined in the CMAH; Statham et al., 2018) for assessing stability and frequency (e.g., SCO, SWE). These were subsequently mapped
to EAWS-classes based on the correspondence proposed by Müller et al. (2016). In Livigno (LIV), the Matrix was applied operationally, but the published forecasts followed North American terminology for describing the sensitivity to triggers. Additionally, some services implemented tools that allowed forecasters to express tendencies within a class. These included services using the ALBINA forecasting software platform (Mitterer et al., 2018), as used for instance in BOZ, TIR, and TRE, as well as the Swiss forecasting software (SAFE, Winkler et al., 2024). In ALBINA, tendencies within classes can be expressed via
sliders allowing near-continuous values between 0 and 100 for snowpack stability, frequency of the lowest snowpack stability class, and avalanche size (Figure 2a). Class boundaries are clearly defined – for example, *poor* corresponds to values from 50 to 74, and *very poor* to values from 75 to 100 – allowing unambiguous conversion into the coarser EAWS class labels used in public communication (see Table 2). In Switzerland, forecasters indicated finer distinctions using modifiers such as *minus*





**Table 1.** Data overview. Warning services are ordered according to the groups they were assigned to (see also Sect. 4.2 ).

| Country | Warning service | | N cases (N days) Dry-snow | N cases (N days) Wet- / gliding snow | Matrix used | Group Dry-snow | Group Wet- / gliding snow |
|---|---|---|---|---|---|---|---|
| Austria | Oberösterreich | OBE | 394 (282) | 170 (141) | yes | A | D |
| Austria | Vorarlberg | VOR | 675 (391) | 255 (168) | yes | A | D |
| Italy | Lombardia | LOM | 759 (147) | 311 (90) | yes | A | D |
| Italy | Val d'Aosta | VDA | 267 (132) | 48 (42) | yes | A | D |
| Italy | Meteomont[a] | MMT | 5290 (263) | 2718 (249) | yes | A | D |
| Slovakia | Slovakia | SVK | 300 (201) | 168 (116) | yes | A | D |
| Austria | Steiermark | STE | 605 (311) | 248 (161) | yes | A | E |
| Italy | Marche | MAR | 214 (80) | 203 (96) | yes | A | E |
| Italy | Trentino | TRE | 549 (336) | 181 (110) | yes | A | E |
| Italy | Veneto | VEN | 384 (121) | 155 (47) | yes | A | E |
| Spain | Catalunya | CAT | 151 (39) | 136 (31) | yes | A | E |
| Germany | Bayern | BAY | 500 (281) | 354 (220) | yes | A | F |
| Norway | Norway[a] | NOR | 10316 (543) | 3995 (355) | yes | A | F |
| Italy | Friuli Venezia Giulia | FRI | 314 (104) | 118 (39) | yes | B | D |
| Austria | Niederösterreich | NIE | 389 (247) | 120 (96) | yes | B | E |
| Austria | Salzburg | SAL | 802 (382) | 398 (221) | yes | B | F |
| Austria | Tirol | TIR | 1040 (370) | 402 (202) | yes | B | F |
| Italy | Bozen–Südtirol[b] | BOZ | 594 (318) | 152 (99) | yes | B | F |
| Spain | Val d'Aran | VAR | 411 (304) | 249 (220) | yes | B | F |
| Italy | Piemonte | PIE | 793 (159) | 111 (31) | yes | C | D |
| Andorra | Andorra | AND | 266 (169) | 201 (158) | yes | C | E |
| Sweden | Sweden | SWE | 1419 (412) | 133 (71) | yes | C | E |
| Austria | Kärnten | KAE | 627 (272) | 239 (125) | yes | C | F |
| Italy | Livigno | LIV | 318 (244) | 157 (119) | yes | C | F |
| Switzerland | Switzerland | SWI | 3876 (423) | 431 (169) | no | C | F |
| Great Britain | Scotland | SCO | 1126 (348) | – | yes | C | |

[a] - not possible to filter unique cases as described in text, [b] - Bozen–Südtirol/Bolzano–Alto Adige

*(-)*, *neutral (=)*, or *plus (+)* (e.g., *poor-*, *poor=*, *poor+*), or selected intermediate labels straddling two adjacent classes (e.g.,
*fair/poor*) (see Figure 2b; Techel et al., 2024). In SWI, factor assessments are used for internal purposes only. For the purpose
of this study, modifier-based entries (e.g., *poor+*) were aggregated into their primary class (*poor*), and intermediate labels (e.g.,
*fair/poor*) were randomly assigned to either the lower (*fair*) or higher (*poor*) class.





**Table 2.** Forecast parameters and their possible values. For brief descriptions, see Tables A1-A3 in Appendix, for full definitions, refer to (EAWS, 2025a, c).

| Forecast parameter | Possible values |
| --- | --- |
| Avalanche problem | new snow, wind slab, persistent weak layer, no distinct problem, * wet snow gliding snow |
| Snowpack stability | very poor, poor, fair, good |
| Frequency of snowpack stability | many, some, a few, (nearly) none |
| Avalanche size | 5 - extremely large, 4 - very large, 3 - large, 2 - medium, 1 - small |

* - These avalanche problems are analysed together as problems relating to dry-snow conditions.

**Individual vs. group forecasting:** In most warning services, a single forecaster was responsible for the forecast for either the entire domain or a specific part of the domain containing one or several sub-regions (e.g., in NOR or SCO), sometimes supported by a second forecaster for review (e.g. TIR). In contrast, SWI followed a fundamentally different approach: the forecasting process always involved at least two (but up to four) forecasters who independently prepared full forecast drafts for the entire forecast domain (referred to as *suggestions*). These were then consolidated in a group discussion prior to publication. For this study, we derived the median factor assessment from the individual suggestions and linked these to the issued danger level resulting from the group consensus.

To harmonize data across services, we applied the following procedure: For each day and micro-region (i.e. smallest geographical entity), we selected the decisive avalanche problem – typically the first listed in the bulletin or the one associated with the highest danger level. If neither criterion was available, the first entry was retained. This step was performed separately for dry-snow problems and for wet- or gliding-snow problems. After selecting the decisive problem, we retained unique entries defined by the combination of date, issuing warning service and/or forecaster, avalanche problem, danger level, and the associated Matrix factors. If the same combination appeared in multiple micro-regions on the same day within the domain of a service, only one instance was retained. This approach was feasible for all services except NOR and MMT, where the data structure did not permit such filtering.

## 4 Methods

The analysis was conducted in three main steps. First, we examined Matrix usage in aggregate, without distinguishing between individual warning services or specific avalanche problems. This allowed us to evaluate whether factor combinations were consistently linked to specific danger levels and whether some Matrix cells showed greater ambiguity or divergence. In the second step, we grouped warning services by shared operational characteristics and analyzed differences in Matrix usage across two avalanche problem categories: (i) *dry-snow* problems, including *new snow*, *wind slab*, *persistent weak layer*, and *no distinct avalanche problem*; and (ii) *wet-snow* and *gliding-snow* problems (EAWS, 2025c). This stratification enabled us



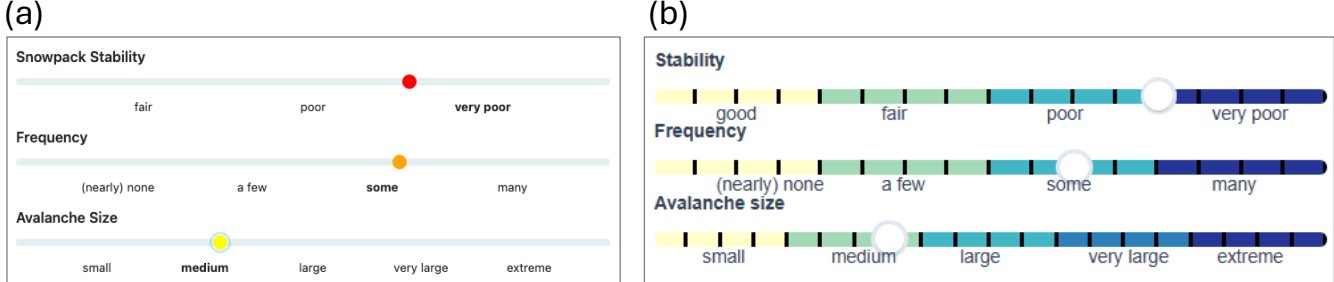

**Figure 2.** Examples of the user interface for assessing snowpack stability, frequency of snowpack stability, and expected avalanche size in the operational forecasting software used by some European warning services. Sliders allow forecasters to indicate tendencies within each factor class. Panel (a) shows the implementation in ALBINA (e.g., BOZ, TIR, TRE) with near-continuous sliders; panel (b) shows the interface used within SAFE in Switzerland (SWI) with underlying modifier-based selection.

to investigate whether observed patterns were consistent across services and groups of avalanche problems, or whether they reflected local practices, forecaster compliance with the Matrix, or differing conceptual models. Finally, we took advantage of the finer-granularity assessments of input factors available in some warning services (SWI and BOZ, TIR, TRE) and explored whether tendencies within classes helped explain variations in danger level assignments within specific Matrix cells.

### 4.1 Degree of Matrix compliance

To investigate the degree of compliance with the Matrix, we derived a binary variable, `disagreement`, by systematically comparing the danger level suggested by the Matrix with the danger level issued in the forecast ($D^{\text{fx}}$). For each assessment instance, the three variables – snowpack stability, frequency of snowpack stability, and avalanche size – were used as inputs to the EAWS-Matrix to determine the Matrix-suggested danger level, specifically the first listed value ($D^1$), resulting from strict application of the Matrix:

$$\texttt{disagreement} = \begin{cases} 1, & \text{if } D^1 \neq D^{\text{fx}} \\ 0, & \text{if } D^1 = D^{\text{fx}} \end{cases}$$

We then computed the proportion of disagreement, $P_{\text{disagree}}$, between the forecast and Matrix-derived danger levels across different danger levels, avalanche problems, and warning services as:

$$P_{\text{disagree}} = \text{mean}(\texttt{disagreement}) \tag{1}$$

### 4.2 Grouping of warning services

We grouped warning services to simplify analysis and presentation while retaining sufficient detail to examine key differences in how the Matrix was applied. This grouping also increased the size of the resulting data subsets, enabling more statistically robust findings. To preserve meaningful distinctions between services, we applied two grouping strategies:




First, for dry-snow avalanche problems, we grouped services based on their degree of Matrix compliance (Section 4.1). We defined three groups using the 50th and 75th percentiles of the disagreement rate ($P_{\text{disagree}}$) as thresholds (see Section 5.2.1, and the *dry-snow* column in Table 1). For the purpose of analysis, we focused on the two groups with the highest and lowest compliance rates:

- (A) Operational use with greater Matrix compliance ($P_{\text{disagree}} < 0.04$), and

- (C) Operational use with lower Matrix compliance ($P_{\text{disagree}} \geq 0.07$) and/or Matrix not used.

Second, for wet- and gliding-snow avalanche problems, we grouped services based on their use of the *very poor* stability class, which varied considerably between warning services (see Section 5.2.2). We split services using the 33rd and 67th percentiles of the proportion of *very poor* classifications (see the *wet-/gliding-snow* column in Table 1), and focused on the groups with the lowest and highest usage of *very poor* stability:

- (D) Proportion of *very poor* stability $P_{\text{stab: very poor}} < 0.25$, and

- (F) Proportion of *very poor* stability $P_{\text{stab: very poor}} \geq 0.56$.

Groups B and E, representing intermediate levels of Matrix compliance and intermediate use of the *very poor* stability class, respectively, were excluded from the main analysis to maintain focus on key contrasts. However, we present their results in the Appendix for completeness.

Since the Scottish data (SCO) did not include avalanche problems, we assumed that most cases represented dry-snow conditions.

## 4.3 Matrix cell usage

For all analyses – whether based on the full dataset or stratified by group and avalanche problem – we first derived usage statistics for each warning service listed in Table 1. For each service, and for each danger level separately, we calculated the proportion of cases in which each Matrix cell was used. If a given danger level was issued in $\leq 3$ cases during the study period, we excluded that level from the cell usage analysis for the corresponding service. We then averaged and normalized these proportions either across all warning services or within the relevant groupings (by warning service group and by avalanche problem type: dry-snow or wet-/glide-snow).

## 4.4 Detecting patterns within Matrix cells

To better understand how individual factor combinations relate to the assigned danger levels – and whether tendencies within a factor class toward higher or lower classes correspond to more frequent use of alternative danger levels – we leveraged the finer-grained assessments of the three Matrix factors available from SWI and the warning services in BOZ, TIR, TRE using the ALBINA software from the 2024-2025 forecast season. Focusing on the most frequently used factor combination *poor-some-size 2* for dry-snow avalanche problems (Figure D1), we identified combinations of sub-classes, which most often predicted





one of the two danger levels as shown in the Matrix. Beside presenting the respective data, we derived decision boundaries applying Classification and Regression Trees (CART, Breiman et al., 2017), as they are well suited for detecting decision

patterns in multi-factor ordinal data. A detailed description of this approach can be found in the Appendix Section B.

## 5 Results

### 5.1 Overall characterization of danger levels with the Matrix

We first analyzed Matrix usage without distinguishing between levels of forecaster compliance or specific avalanche problems. Figure 3 shows how different combinations of stability, frequency, and avalanche size were associated with the assessed danger

levels. Each tile represents a specific Matrix combination, with color intensity indicating the proportion of cases in which that cell was used for a particular danger level.

At danger level 1 (low), only 17% of cases were described using *very poor* stability, suggesting that natural avalanches were rarely considered. The combined proportion of cases classified as *very poor* or *poor* was 44%. According to the intended logic of the Matrix, *fair* stability should only have been selected if both *very poor* and *poor* stability were assessed as *none*

*or nearly none* in the region. Thus, the data implied that neither natural avalanche activity nor human-triggered avalanches were considered relevant in more than half of the cases (56%). Avalanche size was typically classified as size 1, with a smaller share (20%) assigned to size 2. It is important to note that not all warning services assessed Matrix factors when no avalanche problem was identified. Consequently, the proportions shown in Figure 3 did not equally reflect all services. For instance, for SWE, 39% of the danger level 1 forecasts lacked factor estimates.

At level 2 (moderate), usage was concentrated in the *poor* stability panel (62%), which is typically associated with human-triggered avalanches. In 15% of forecasts, stability was described as *very poor*, most often in connection with avalanche size 2. The frequency of locations with *very poor* or *poor* stability was most commonly assessed as *a few* or *some*, while 17% of cases would have been classified as *none or nearly none* for these two stability classes—corresponding to situations where stability was assessed as *fair*. Avalanche size was predominantly size 2, accounting for 87% of cases.

At level 3 (considerable), stability was equally often described as *very poor* and *poor*, with frequency most commonly assessed as *some* (82%). Avalanche size was typically size 2 or size 3. Notably, size 3 was more frequently associated with *poor* stability, while size 2 occurred more often with *very poor* stability.

At level 4 (high), there was a strong concentration in the cell *very poor–many–size 3*, which was used in 60% of cases. Nonetheless, in 16% of the cases, forecasters chose *poor* stability in combination with *many* locations and avalanche size 3 or

4 when issuing danger level 4. These situations are often referred to as "skier-level 4" (e.g., SLF, 2024).

Danger level 5 (very high) was issued only once during the study period and is therefore not shown in Figure 3; in this case, it was described as *very poor–many–size 4*.

Across all danger levels, most Matrix cells were predominantly used for a single danger level, with a few important exceptions. The cell *poor–some–size 2* was the most frequently used combination for level 2 (moderate) (31%), but also accounted




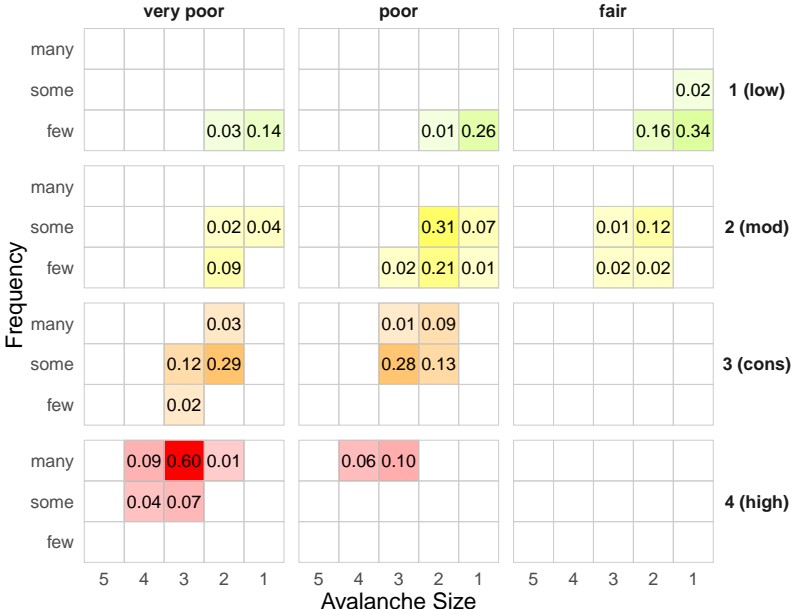

**Figure 3.** Matrix use by danger level: Proportion of cases that a specific danger level (rows from top: $D = 1$ (low) to $D = 4$ (high)) was used for a specific matrix combination. Colour shading corresponds to the proportion of cases. Cells with less than 1% usage are not shown.

for 13% of level 3 (considerable) assessments. Similarly, the cell *very poor–some–size 3* was used in 12% of level 3 cases and 7% of level 4 cases.

## 5.2 Avalanche-problem specific characterization of danger levels with the Matrix

We next analyzed Matrix usage separately for dry-snow and for wet- or glide-snow avalanche problems. The corresponding data contributions from each warning service are summarized in Table 1.

### 5.2.1 Dry-snow avalanche problems

Comparing the issued danger level with the Matrix-suggested danger level showed that nearly all warning services occasionally deviated from the Matrix (median $P_{\mathrm{disagree}} = 4\%$; Fig. 4a). The highest disagreement rates were observed in Andorra (AND), Scotland (SCO), and Switzerland (SWI), where deviations occurred in 36–39% of cases. Piemonte (PIE), Livigno (LIV), and Sweden (SWE) also showed comparably frequent deviations (10–21%), while in half of the warning services, deviations were relatively rare ($P_{\mathrm{disagree}} < 4\%$).

Figure 5 illustrates how combinations of stability, frequency, and avalanche size were associated with the issued danger levels for dry-snow avalanche problems. Results are shown by the groups introduced in Figure 4a. Group A – characterized by higher compliance with the Matrix – exhibited a clear and consistent link to the Matrix structure. Most Matrix cells were associated with a single danger level, and overlaps were rare: only three factor combinations – *fair–a few–size 1*, *poor–some–size 2*, and



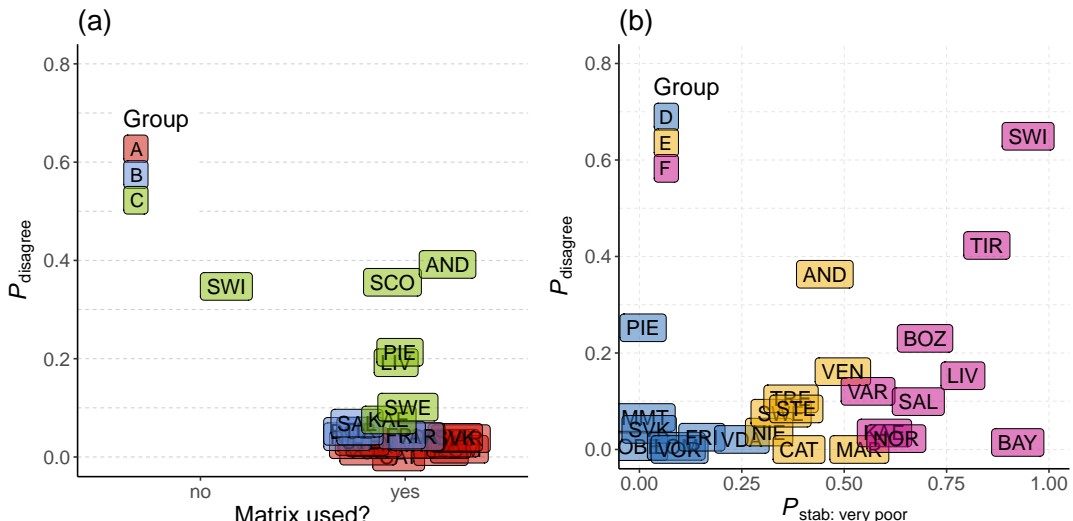

**Figure 4.** Proportion of disagreement between Matrix-suggested and issued danger level ($P_{\mathrm{disagree}}$) for (a) dry-snow avalanche problems and (b) wet- and glide-snow avalanche problems. In (a), use of the Matrix is shown on the x-axis, in (b) the proportion of *very poor* stability assessments ($P_{\mathrm{stab:\ very\ poor}}$). Each point represents one warning service, colored by group as defined in Section 4.2.

*very poor–some–size 3* – were used in more than 1% of cases for an alternative danger level. In contrast, Group C – defined by $P_{\mathrm{disagree}} \geq 7\%$, though presumably characterized by high heterogeneity in terms of Matrix integration in the forecasting process – showed a much broader distribution of danger levels across Matrix cells, indicating greater overlap in the use of the same factor combinations for different danger levels.

     Two factor combinations stood out for both groups: *poor–some–size 2* and *very poor–some–size 3*. These combinations were

among those where deviations from the Matrix-suggested danger level were more frequent, even among Group A forecasters.

     According to the Matrix (Fig. 1), the primary danger level for *poor–some–size 2* is 2 (moderate), with 3 (considerable) as the second option. While Group C warning services most often used this combination to describe 3 (considerable), they also used it frequently for 2 (moderate). In contrast, Group A primarily used neighboring Matrix cells when issuing 3 (considerable). Nonetheless, even in Group A, 7% of 3 (considerable) assessments fell into this combination. This cell thus appeared to serve

as a key transition zone between danger levels 2 and 3, where expert judgment diverged based on the level of compliance with the Matrix.

     A second notable difference between groups involved the cell *very poor–some–size 3*, where the Matrix suggests 3 (considerable) as the primary level and 4 (high) as the secondary. Group C used this combination for 4 (high) in 25% of cases, while it was less frequently used for 3 (considerable) (10%). In contrast, Group A used this combination in 11% of cases for 3

(considerable), and only 3% of cases for 4 (high).





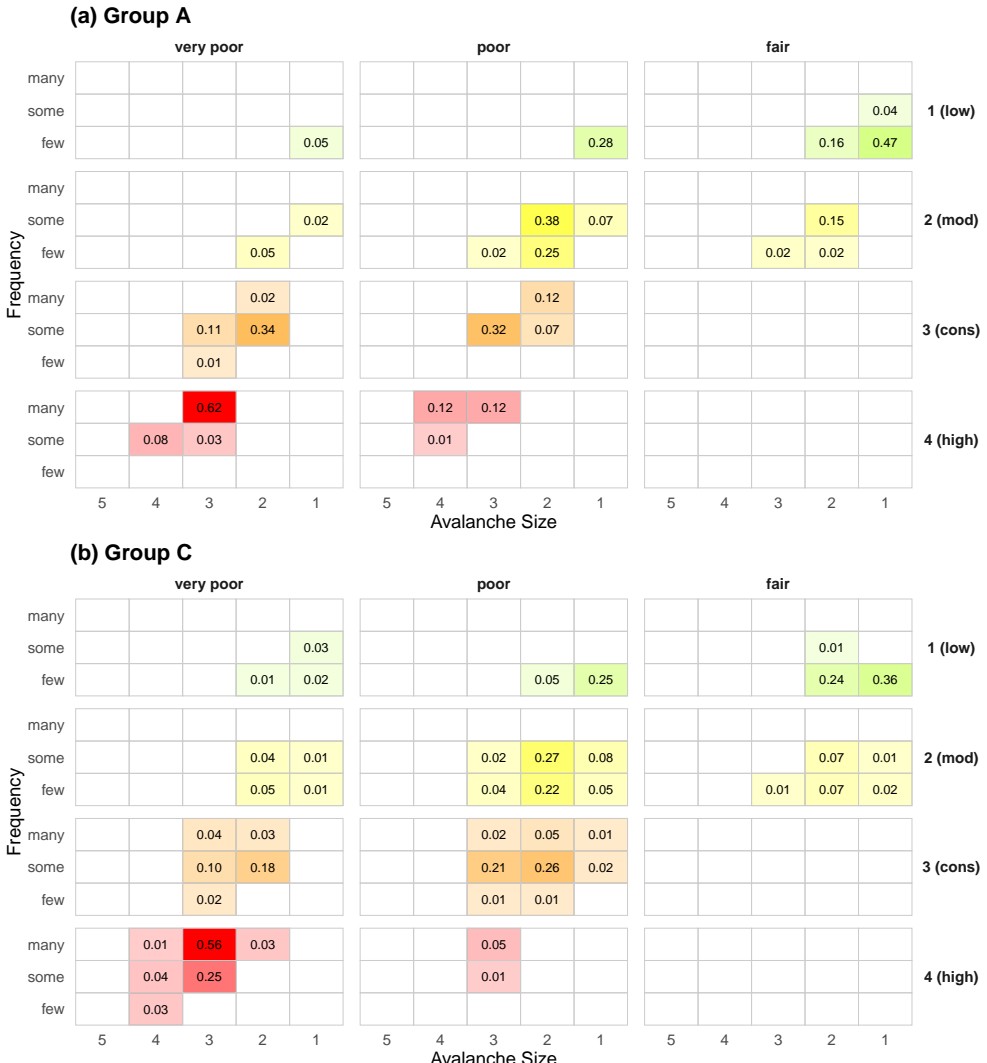

**Figure 5. Matrix use for dry-snow avalanche problems:** Proportion of cases that a specific danger level (rows from top: $D = 1(\text{low})$ to $D = 4(\text{high})$) was used for a specific matrix combination. Colour shading corresponds to the proportion of cases. Cells with less than 1% usage are not shown. Group A forecasters rarely deviated from the Matrix-suggested danger level, group C forecasters deviated $\geq 7\%$ of the time. The respective figure for warning services lying in between these two (group B) is shown in the Figure C1 in the Appendix. Absolute use of Matrix cells, regardless of danger level, is shown in Figure D1 in the Appendix.

Despite these differences, both groups shared several patterns. For 1 (low), the most frequently used combinations were *fair–a few–size 1* or *poor–a few–size 1*. When issuing 2 (moderate), the typical combinations were *poor–some–size 2* or *poor–a few–size 2*. And when issuing 4 (high), both groups predominantly used the factor combination *very poor-many-size 3*.



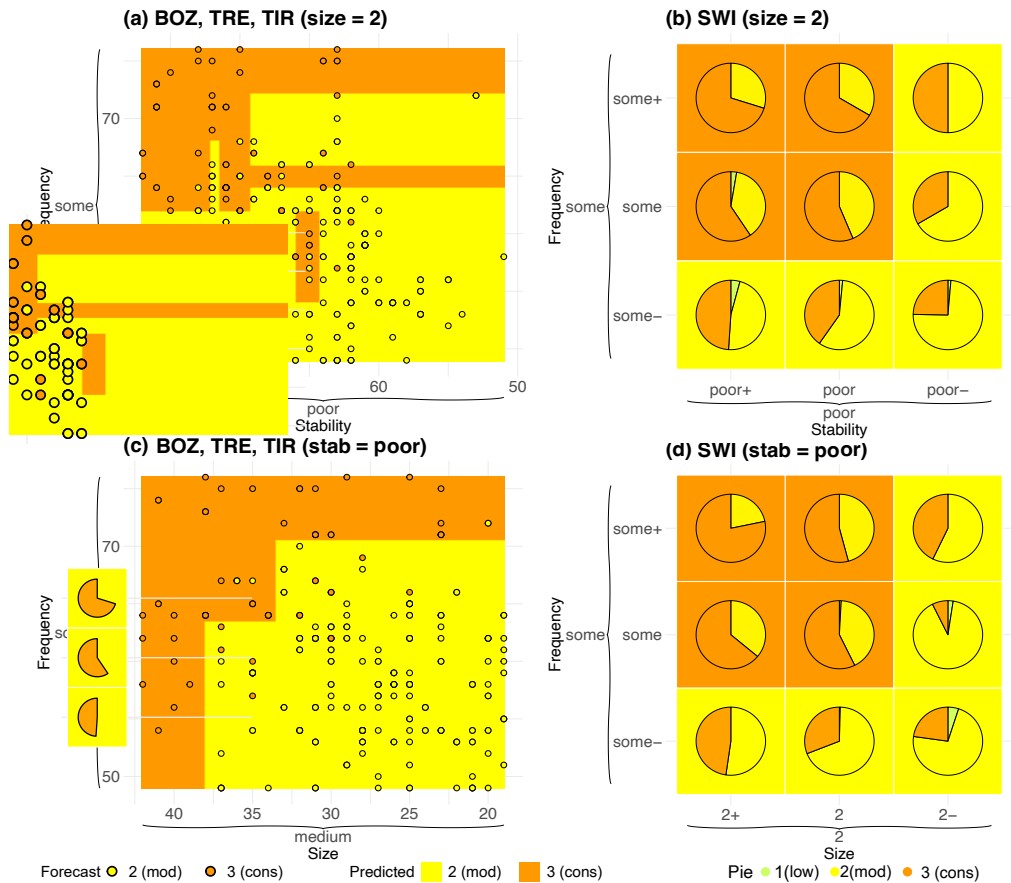

**Figure 6.** Distribution of issued danger levels as a function of factor combinations for different warning services, illustrating tendencies within coarse Matrix categories for the Matrix cell *poor-some-size 2*. Panels (a, b) show stability and frequency (with avalanche size fixed to 2), while panels (c, d) show frequency and size (with stability fixed to *poor*). The left column (a, c) presents data from BOZ, TIR, and TRE, which use the ALBINA forecasting software (see also Figure 2); the right column (b, d) shows data from SWI. Forecasted danger levels are indicated by circles (a, c) or pie segments (b, d). Background colors reflect the predicted danger level based on the CART model.

Although the Matrix required forecasters to classify conditions into a small number of discrete categories, the underlying processes are inherently continuous. In BOZ, TIR, and TRE (using the ALBINA software), as well as in SWI (using the SAFE software), forecasters were able to assess the Matrix factors with finer granularity (Figure 2). This allowed a closer examination of how subtle differences in factor assessments translated into differences in the assigned danger level. We focused on the combination *poor–some–size 2*, the most frequently used Matrix cell (Figure D1), which also showed considerable variation in the assigned danger level (Figure 5) – either 2 (moderate), as suggested by the Matrix, or 3 (considerable).

For the three warning services BOZ, TIR, TRE (using ALBINA) and for SWI separately, Figure 6 shows how the assigned danger levels varied within this cell. Panels (a) and (b) display the distribution of danger levels as a function of snowpack





stability and frequency, while avalanche size was fixed to *size 2*. Panels (c) and (d) focus on cases where stability was classified as *poor*, showing how danger levels varied as a function of frequency and size – corresponding directly to the Matrix cell shown in Figure 1.

In SWI, the issued danger levels were either 2 (moderate) (42%) or 3 (considerable) (57%), with only 12 cases (1% of 964) assigned level 1 (low). In contrast, the ALBINA services issued level 2 in 75% of 282 cases and level 3 in 25%. Despite this difference in the relative preference for level 2 versus level 3, similar tendencies emerged across groups. Slider positions leaning toward *very poor* stability and *many* locations were more often associated with level 3 (considerable) (Figure 6a, b), whereas shifts toward *fair* stability and a *few* locations tended to result in level 2 (moderate). The services BOZ, TIR, TRE

showed a pronounced diagonal pattern in slider use, highlighting the interplay between stability and frequency. In SWI, the proportion of level 2 (moderate) assignments decreased systematically with decreasing stability and increasing frequency. A similar trend was observed for the combination of frequency and size (Figure 6c, d): danger level 3 (considerable) was more likely when both factors increased. The CART predictions shown in the figure backgrounds closely mirrored these tendencies. While the prediction surfaces were generally well defined, some scatter was present (notably in Figure 6a).

### 300   5.2.2   Wet-snow and glide-snow problems

Figure 4b illustrated the use of *very poor* stability to describe wet-snow or glide-snow conditions and the proportion of disagreements between the issued danger level and the primary danger level shown in the Matrix. While forecasters in several services (notably SWI, TIR, and BAY) predominantly assessed wet-snow and glide-snow avalanche problems using *very poor* stability, a few services never used this stability class. This was striking, given that wet-snow and glide-snow avalanches are

typically associated with natural avalanche release (Schweizer et al., 2020), and are thus conceptually linked to *very poor* stability (Table A1). This divergence in interpretation was also reflected in the degree of disagreement with the Matrix-suggested danger level. Services that used *very poor* stability more frequently also tended to show greater disagreement (Spearman $\rho = 0.38$, $p < 0.1$), and vice versa. Notable exceptions included BAY, where stability was described as *very poor* in most situations (92%) while high agreement with the Matrix was maintained (99%), and Piemonte (PIE), where $P_{\mathrm{disagree}}$ was high

(25%) even though stability was essentially never assessed as *very poor* (1%). In the case of BAY, forecasters typically assigned smaller avalanche sizes to the same danger levels compared to, for instance, SWI and TIR, who showed much lower agreement with the Matrix-suggested danger levels. For example, at 1 (low), 93% of BAY cases were size 1, compared to 37% in SWI and 13% in TIR. Overall, for wet-snow and glide-snow problems, the median disagreement rate $P_{\mathrm{disagree}}$ was 7%, slightly higher than the 4% observed for dry-snow avalanche problems.

These differences in stability assessments and levels of compliance with the Matrix-suggested danger level carried over into the final danger level assignments, leading to a broad range of factor combinations and considerable variation in how the same danger level was described. Splitting the warning services into three groups (Section 4.2, Figure 4b) based on their use of *very poor* stability, and focusing on the two groups that either used this class infrequently (group D, $P_{\mathrm{stab:\ very\ poor}} < 0.25$) or often (group F, $P_{\mathrm{stab:\ very\ poor}} \geq 0.56$), revealed substantial differences in Matrix cell usage (Figure 7). Group D forecasters

most often used *poor* and *fair* stability across danger levels 1 (low) to 3 (considerable). Even for 3 (considerable), 62% of



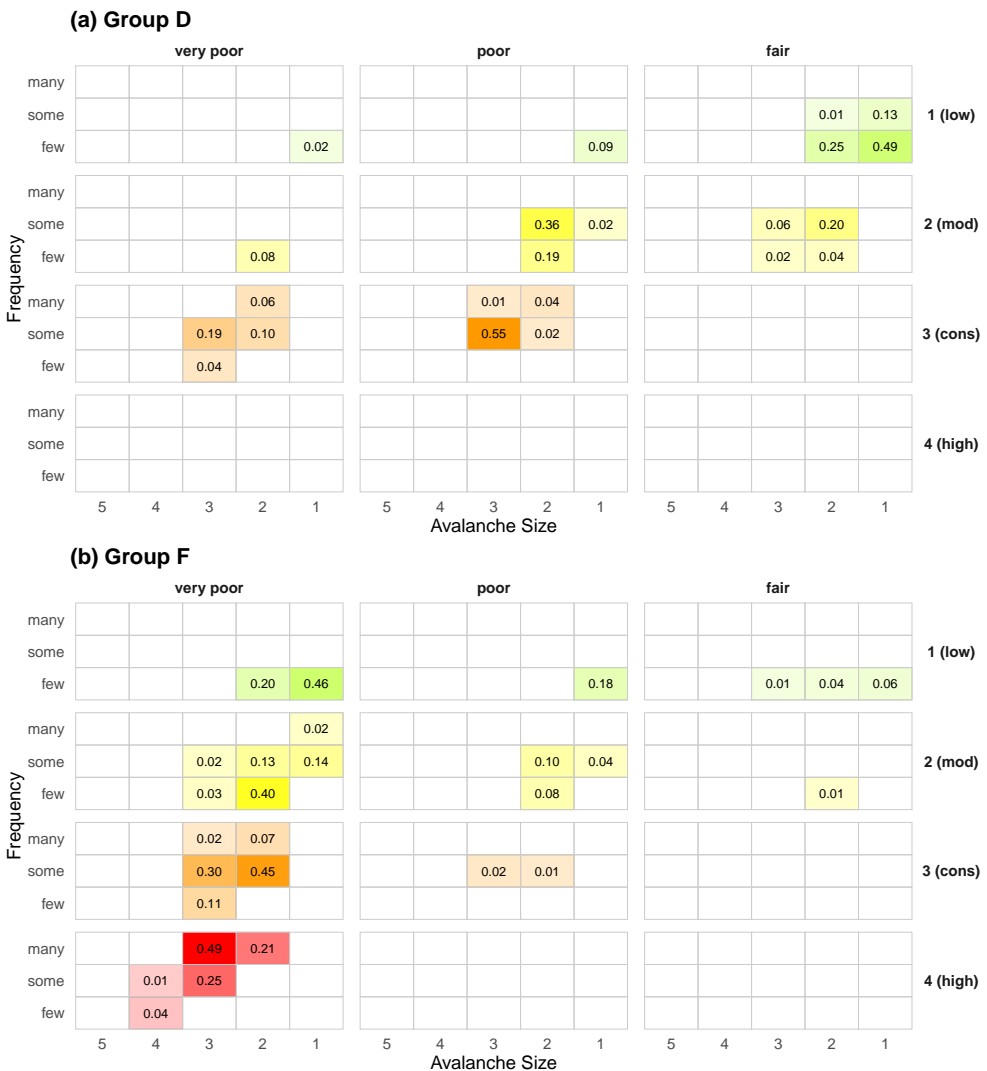

**Figure 7. Wet-snow and glide-snow problems:** Proportion of cases that a specific danger level (rows: $D = 1(\text{low})$ to $D = 4(\text{high})$) was used for a specific matrix combination. Colour shading corresponds to the proportion of cases. Cells with less than 1% usage are not shown. Group D forecasters used very poor stability $< 25\%$ of the time, group F forecasters $\geq 56\%$ of the time (see Figure 4b). The respective figure for warning services lying in between these two (group E) is shown in the Figure C2 in the Appendix. Absolute use of Matrix cells, regardless of danger level, is shown in Figure D1 in the Appendix.

cases involved *poor* stability. In contrast, group F forecasters predominantly used the *very poor* stability panel, even when describing 1 (low), where this occurred in 66% of cases. Furthermore, group D forecasters did not use danger level 4 (high) at all when assessing wet- or glide-snow problems. As a result, similarities between these groups related primarily to avalanche





size, which increased with rising danger level in both groups – from size 1 at 1 (low), to size 2 at 2 (moderate), and size 3 at 3
(considerable) in group D and size 2 to size 3 in group F.

## 6 Discussion

### 6.1 Interpretation of findings

The revised EAWS Matrix was developed through expert consensus in a non-operational setting, where factor combinations
were linked to danger levels without reference to specific forecasting situations and without the time pressure or emotional
demands of real-world forecasting (Müller et al., 2025). This study represents the first systematic analysis of how the Matrix
performed under operational conditions – where forecasters had to assess factors and determine danger levels under time
constraints, uncertainty, and the potential for serious consequences if forecasts were incorrect. While Matrix usage was broadly
consistent with its design, our findings revealed key patterns and variations that offer important insights for refinement and
further harmonization.

Importantly, these conclusions must be viewed in light of a fundamental limitation: the EAWS Matrix ultimately rests on
expert judgment – both in its original development (Müller et al., 2025) and in its operational application. Since avalanche
danger and its defining factors cannot be measured objectively, there is no external reference for validating how the Matrix is
used. This introduces a degree of circularity into our evaluation: most forecasters assess factors and danger levels through the
lens of the Matrix, and the Matrix itself was built on expert interpretation of those same factors. In particular, services with
greater compliance may produce assessments that primarily reflect the Matrix logic. As a result, it is difficult to determine
whether deviations from the Matrix reflect inconsistent application or justified adaptations to specific avalanche conditions.
Prominent patterns that emerge across services – regardless of the level of Matrix compliance – may therefore offer particularly
valuable insights for refining the Matrix.

Despite this limitation, our analyses showed that most danger levels were most frequently associated with a consistent
combination of stability, frequency, and size. This pattern was particularly evident for danger levels 1 (low), 2 (moderate), and
4 (high), suggesting that the Matrix provides a robust and interpretable structure. Notably, certain cells that included a second
danger level ($D^2$) were almost always assigned a single level during operational forecasting – even by Group C forecasters
with lower Matrix compliance. This indicates a broad consensus on the appropriate level in these cases. Such cells could likely
be simplified by removing $D^2$ without significant loss of guidance.

350 A small number of cells – most notably *poor–some–size 2* and *very poor–some–size 3* – acted as transition zones, showing
frequent overlap between adjacent danger levels and varying importance across levels and forecaster groups. For danger levels
3 (considerable) and 4 (high), these cells were among the most frequently used by Group C forecasters, even though these
levels were indicated only as secondary danger levels in the Matrix. The reasons for this discrepancy remain speculative.
Divergent use may stem from inconsistencies in interpreting data or from differing interpretations of the factor classes (Techel
355 et al., 2024), particularly the frequency categories, or – also speculative – from a tendency among Group A forecasters to avoid



Matrix cells that would not produce the intended danger level. Notably, even Group A forecasters occasionally selected these cells issuing $D^2$.

The exemplary analysis of finer-granularity factor assessments for the combination *poor–some–size 2* (Figure 6) highlighted a key limitation of the Matrix: reducing avalanche conditions to a small number of discrete classes sacrifices nuance. Variability and uncertainty in factor estimation cannot be captured within coarse categories and are therefore lost in subsequent analysis. Nonetheless, the differences between the four investigated warning services were relatively minor even though this cell was used dominantly for 3 (considerable) in SWI, and 2 (moderate) by BOZ, TIR, and TRE. These variations likely reflect both the extent to which danger level assignment was anchored in the Matrix (e.g., BOZ, TIR, TRE) and subtle differences in the interpretation of factor classes. Tendencies to favor either level 2 (moderate) or level 3 (considerable) based on the location of factor assessments within the cell were broadly comparable. In SWI, where the Matrix is not used, a few sub-factor combinations spanned up to three danger levels. Still, forecasts issued by SWI and the ALBINA services (BOZ, TIR, TRE) often converged on similar danger levels for comparable (sub-class) settings. This agreement—despite procedural and interpretational differences – provides evidence of convergent validity (Campbell and Fiske, 1959): the idea that different methods aiming to assess the same concept should yield similar results. In this case, both Matrix-based and independently derived forecasts point to a shared conceptual understanding of avalanche danger. This suggests that – even without external reference data – the observed convergence supports both the conceptual soundness of the Matrix and the feasibility of harmonized interpretation across services.

Fully understanding these differences would require additional context – such as forecaster rationale or detailed knowledge about regional snow and weather conditions. While finer granularity helps preserve variation within pre-selected coarser classes, the appropriate level of granularity remains a topic for discussion. It is well known that humans can typically assess only a small number of classes reliably (e.g., Miller, 1956). Therefore, combining absolute and relative assessments (e.g., Kahneman et al., 2021) – in our case, first assessing the discrete Matrix input classes and then using a relative ranking within them, generally provides more reliable estimates. For instance, in SWI, where several forecasters independently prepare a forecast draft in parallel, they selected the same sub-class (as shown in Figure 2b) in 55% of cases and were within two neighboring sub-classes 90% of the time (Techel, F., unpublished data). This highlights that reasonably reliable estimates are possible even with a higher-resolved absolute-relative scale, and that forecasters' estimated tendencies are often mirrored by their peers.

Some cells remain rarely used across all services (Figures D1 and D2). This lack of empirical support reduces confidence in the danger levels suggested in the Matrix and highlights that these assignments still rely solely on expert elicitation when developing the Matrix (Müller et al., 2025).

The stratified analyses (Sect.5.2.1) confirmed that forecaster compliance influenced Matrix usage. Group C services, which disagreed more frequently with the danger level proposed in the Matrix, showed broader distributions and more frequent use of alternative danger levels than services with more standardized use (Group A). Interestingly, even some Group A and B services – generally in strong alignment with the Matrix for dry-snow conditions – showed comparably large deviations when assessing wet- and glide-snow conditions (Figure4). Transparent documentation of such deviations remains essential and may provide important input for community-wide discussion of these cells.



A main objective of integrating the Matrix into the forecast process is to improve consistency. Our findings suggest that services with high Matrix compliance (Group A) indeed used a narrower set of Matrix cells for each danger level, indicating greater consistency in how factor combinations were linked to the danger level. This suggests the Matrix may help standardize assessments. However, consistency in output does not necessarily imply consistency or accuracy in interpretation. It is pos-

sible – though speculative – that forecasters sometimes adjusted input factors to reach a desired Matrix outcome, rather than exclusively letting the data guide factor selection. In addition, each factor estimate carries uncertainty, depending on available data and the forecaster's ability to retrieve and correctly interpret it (Stewart and Lusk, 1994), with potentially adverse effects when inputting these into the Matrix. To illustrate this, imagine two forecasters assessing identical conditions. If their estimates differ by only one neighboring class for a single factor in half the cases, while fully agreeing otherwise, the resulting danger

level would differ 21% of the time when all Matrix combinations are considered once. As only one forecast can be considered "correct", such inconsistencies inevitably reduce accuracy (Techel et al., 2024). Without access to external validation data or forecaster rationale, it remains difficult to determine whether the Matrix promotes genuinely consistent (factor) interpretations or merely uniform outputs.

Finally, while Matrix use was relatively consistent for dry-snow avalanche problems, forecasts for wet- and glide-snow

problems showed considerably more variation – both compared to dry-snow assessments and between services. Similar discrepancies have been documented in Canada (Clark and Haegeli, 2018; Clark, 2019), where identical combinations of likelihood of avalanches (the North American counterpart to stability and frequency in the Matrix) and avalanche size resulted in different danger levels depending on avalanche problem type and, at times, the forecasting agency. In our analysis, the most notable differences related to the classification of snowpack stability for wet- and glide-snow avalanche problems. Although

these avalanche types are generally associated with natural avalanche occurrence (Schweizer et al., 2020; Hutter et al., 2021), and thus conceptually with *very poor* stability (Table A1), services showed considerable divergence in their stability ratings (Figure 4b). Again, we can only speculate whether this reflects different conceptual models or deliberate adjustments of factor inputs to achieve expected Matrix outcomes. To promote harmonized application of the Matrix, a clearer, shared framework for assessing stability in wet- and glide-snow contexts is needed.

## 6.2    Recommendations for improving the Matrix and workflow

Based on the findings outlined above, we propose the following recommendations:

– **Simplify cell content.** Certain cells of the Matrix were used quite strongly for single danger levels (Figure 3). These cells include the factor combinations *very poor-a few-size 3* and *very poor-some-size 2*, which were strongly used for 3 (considerable), and *poor-a few-size 2* and *poor-some-size 1* , which were used mostly for 2 (moderate). Therefore these

cells could be reduced to further emphasize one specific danger level without significant loss of information.

– **Use white shading for under-supported cells.** Beyond those cells already marked white in the Matrix (Figure 1), several additional cells were rarely used and could also be shaded white to indicate higher uncertainty, as these assignments still





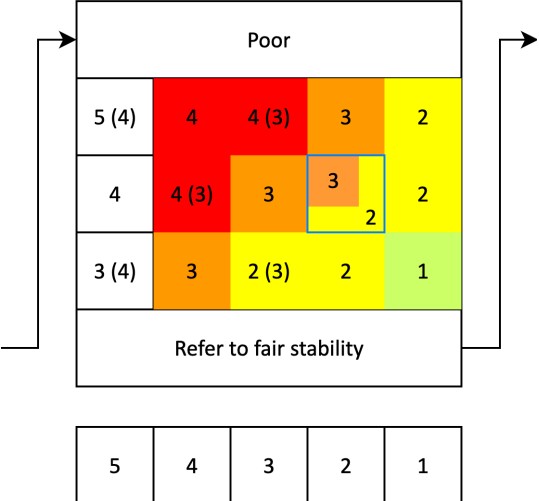

**Figure 8.** Close-up of the Matrix cell *poor-some-size 2*, showing an internal differentiation based on finer-granularity factor assessments, as identified in Figure 6.

rely exclusively on the expert survey underlying the Matrix. These cells are *fair* stability in combination with either avalanche size 4 or 5, or frequency *many*, and *poor* stability in combination with avalanche size 5.

– **Investigate transition zones and promote transparency and additional guidance in ambiguous cases.** The cells *poor–some–size 2* and *very poor–some–size 3* were frequently assigned to two danger levels, highlighting persistent ambiguity in these transition zones. These areas should be further examined to better understand the causes of variation and to refine the corresponding guidance. While the Matrix should be applied in line with agreed standards, these cells remain open to interpretation. As objective validation is not feasible, we recommend that deviations from the primary

danger level ($D^1$) be documented transparently. Clear documentation helps distinguish well-founded expert judgment from inconsistent application and facilitates learning across services. This may include practices already in use – such as indicating tendencies within factor classes, as used and analyzed for four warning services (Figures 2 and6). Moreover, such information could be used to refine the Matrix further. For instance, Matrix cells such as *poor–some–size 2* could be split further to provide more detailed guidance to forecasters (Figure 8).

– **Clarify assessment process for wet- and glide-snow problems.** A harmonized conceptual framework is needed to guide how these avalanche problems are assessed. This may require revised definitions, forecaster training, or updated guidance in the workflow.





## 6.3 Limitations

A key limitation is that we lack precise information about how the Matrix was implemented operationally within each warning
service. While some services likely provided formal internal guidance – for instance by implementing the Matrix directly in
the forecasting software (e.g., ALBINA software) – or training, others allowed for greater forecaster discretion in referring to
the Matrix or did not use the Matrix, but rather reached a final decision following group discussion (SWI). We also do not know
the rationale behind individual forecaster decisions or the situational context in which factor assessments were made. These
missing layers of information limit our ability to distinguish between genuine differences in interpretation and procedural
differences across services —- and would be valuable for understanding when and why forecasters adhered to or deviated from
the Matrix, particularly in ambiguous or transitional cells. Moreover, variations in Matrix use and data collection procedures
described in Section **??** required harmonizing the data across services, which may have introduced an unknown degree of
information loss – including potentially relevant detail. Finally, while grouping warning services facilitated data analysis and
presentation, some groups – such as Group C – were likely heterogeneous with respect to the underlying reasons for deviating
from the Matrix.

## 7 Conclusions

This study provides the first comprehensive assessment of how the revised EAWS-Matrix has been used operationally across
European avalanche warning services. The findings confirm that the Matrix structure is broadly effective, with most cells sup-
porting consistent danger level assignment – even among services with a tendency to diverge from the Matrix-suggested danger
level. However, two factor combinations – *poor–some–size 2* and *very poor–some–size 3* — emerged as areas of ambiguity that
warrant closer examination. Analyzing finer-granularity factor assessments can reveal tendencies within Matrix fields and may
offer a path toward more specific guidance in cells that currently contain two danger levels. Rarely used cells remain a source of
uncertainty and should be visually marked. While Matrix use was relatively consistent for dry-snow avalanche problems, sub-
stantial inconsistencies were observed for wet- and glide-snow problems, especially in the classification of snowpack stability.
This highlights a need for broader harmonization of assessment practices across services. Overall, the results provide guidance
for refining the Matrix and underscore the importance of transparent documentation and shared interpretation frameworks in
domains where expert judgment plays a central role.

Ultimately, the assessment of avalanche danger levels should be guided as directly and transparently as possible by avail-
able data – not by individual forecasting styles, conceptual preferences, or service-specific traditions. Forecasts should reflect
a shared understanding of avalanche danger and its determining factors, rather than the personality or institutional context
of the forecaster. While some degree of expert interpretation is unavoidable, danger level assignment should result from a
forward-looking evaluation of the evidence, not reverse reasoning from a desired outcome. Given the inherent uncertainty and
subjectivity of the data, providing objective and targeted recommendations for improving the Matrix and its use is undoubtedly
challenging but remains essential for producing credible and consistent public avalanche forecasts across Europe.





**Appendix A: Definition of factors**

**Table A1.** Snowpack stability classes referring to the point scale, and the type of triggering typically associated with these classes. For the full table, including typical observations related to each class, see EAWS (2025a, Figures A1-A3).

| Stability class | Description |
|---|---|
| Very poor | very easy to trigger (e.g., natural) |
| Poor | easy to trigger (e.g., a single skier) |
| Fair | difficult to trigger (e.g., explosives) |
| Good | stable conditions |

**Table A2.** Frequency classes of snowpack stability, taken from EAWS (2025a, Table 2).

| Frequency class | Description | Evidence (e.g., observations) |
|---|---|---|
| Many | Points with this stability class are abundant. | Evidence for instability is often easy to find. |
| Some | Points with this stability class are neither many nor a few, but these points typically exist in terrain features with common characteristics (i.e., close to ridgelines, in gullies). | |
| A few | Points with this stability class are rare. While rare, their number is considered relevant for stability assessment. | Evidence for instability is hard to find. |
| None or nearly none | Points with this stability class do not exist, or they are so rare that they are not considered relevant for stability assessment. | |

**Table A3.** Avalanche size classes, taken from EAWS (2025a, Table 3).

| Size class | Label | Destructive potential |
|---|---|---|
| 1 | Small | Unlikely to bury a person, except in run out zones with unfavorable terrain features (e.g., terrain traps). |
| 2 | Medium | May bury, injure, or kill a person. |
| 3 | Large | May bury and destroy cars, damage trucks, destroy small buildings and break a few trees. |
| 4 | Very large | May bury and destroy trucks and trains. May destroy fairly large buildings and small areas of forest. |
| 5 | Extremely large | May devastate the landscape and has catastrophic destructive potential. |




## Appendix B: Details on the CART model for sub-class analysis

Classification and Regression Trees (CART, Breiman et al., 2017) are non-parametric models that recursively partition the predictor space to construct interpretable decision trees for classification or regression. CART models are highly interpretable, as each split corresponds to a simple decision rule. The resulting tree structure provides clear insight into the relationships

between predictors and the response. CART is robust to outliers, can handle both numerical and categorical predictors, and does not require assumptions about the distribution of the data. For this study, we applied the `rpartScore` package (Galimberti et al., 2012), which extends CART to ordinal outcomes by incorporating misclassification costs that reflect the ordered structure of the response variable. Unlike standard CART procedures that treat all misclassifications equally, this approach recognizes that errors between adjacent ordinal categories are less severe than errors between distant categories. The methodology assumes

that numerical scores are assigned to the ordered categories of the response variable, reflecting the inherent ordinal structure of the data.

The dataset consists of $p$ predictors $X = \{X_1, \ldots, X_p\}$ (here: stability, frequency, size) and an ordinal target variable $Y$ (here: $D$). For each observation $i = 1, \ldots, N$, there is a data pair $(y_i, x_i)$, where $y_i$ is the target variable and $x_i = (x_{i1}, \ldots, x_{ip})$ represents the predictor values.

Trees were grown using the *Generalized Gini impurity function*, which incorporates misclassification costs calculated based on the ordinal distances between categories. The splitting criterion selects the predictor variable and split point that maximizes the reduction in the Generalized Gini impurity at each node. For a given node $m$ representing region $R_m$ with $N_m$ observations, the proportion of class $k$ is

$$\hat{p}_{mk} = \frac{1}{N_m} \sum_{x_i \in R_m} I(y_i = k), \quad \hat{p}_{mk} \in [0, 1]$$

where $I(\cdot)$ is the indicator function. The Generalized Gini impurity for node $m$ is then defined as

$$G_m = \sum_{k=1}^{K} \sum_{l=1}^{K} c_{kl} \hat{p}_{mk} \hat{p}_{ml}$$

where $K$ is the number of categories, and $c_{kl}$ is the misclassification cost between categories $k$ and $l$, typically set as the absolute or squared difference between their scores (e.g., $c_{kl} = |k - l|$ or $c_{kl} = (k - l)^2$). This impurity measure accounts for both the frequency of misclassifications and their severity based on the ordinal distance between categories, ensuring that splits are chosen to minimize not only the number but also the seriousness of classification errors in the context of ordinal data.

To avoid overfitting, cost-complexity pruning was applied. Two pruning criteria were evaluated:

- **Total misclassification rate** (`prune = "mr"`): Minimizes the proportion of misclassified observations.

- **Total misclassification cost** (`prune = "mc"`): Minimizes the cumulative cost of misclassifications, weighted by ordinal distance.

The complexity parameter (`cp`) is a central hyperparameter in the `rpartScore` framework that controls the trade-off

between tree complexity and model fit (Breiman et al., 2017; Galimberti et al., 2012). At each split, `cp` specifies the minimum





reduction in the overall cost-complexity measure required for a split to be retained in the tree. As `cp` increases, the algorithm produces smaller, simpler trees by pruning branches that do not sufficiently decrease the impurity, thereby reducing the risk of overfitting. Conversely, a lower `cp` allows for more complex trees, which may capture more structure but risk overfitting the training data.

The analysis employed a comprehensive hyperparameter tuning approach using repeated cross-validation. A 10-fold cross-validation with 3 repetitions was implemented to ensure robust model evaluation. The hyperparameter grid included:

- **Complexity parameter**: Values ranging from 0.01 to 0.3 in increments of 0.05

- **Split functions**: Both absolute (`"abs"`) and squared (`"quad"`) difference approaches

- **Pruning measures**: Both misclassification rate (`"mr"`) and misclassification cost (`"mc"`) criteria

To address potential class imbalance in the ordinal response variable, the Synthetic Minority Oversampling Technique (SMOTE) was applied during the cross-validation process. This technique generates synthetic examples of minority classes to balance the training data while preserving the ordinal structure of the response variable. However, SMOTE was only used in cases where the predictors were continuous. In the case of SWI, where the predictors were categorical, no SMOTE was applied.

Model performance was assessed using Matthews Correlation Coefficient (MCC) as the primary evaluation metric (Matthews, 1975). MCC is well-suited for imbalanced class distributions, as it considers all elements of the confusion matrix and avoids the pitfalls of standard accuracy metrics when one class dominates. In this analysis, MCC is especially appropriate for SWI, where predictors were categorical and SMOTE was not used. Without SMOTE, traditional metrics could overestimate performance, but MCC ensures a fair and robust evaluation even with class imbalance.

$$\text{MCC} = \frac{TP \times TN - FP \times FN}{\sqrt{(TP+FP)(TP+FN)(TN+FP)(TN+FN)}}$$

where $TP, TN, FP$, and $FN$ represent true positives, true negatives, false positives, and false negatives, respectively. While the formula above shows the binary case, for multiclass classification the MCC is computed using a generalized confusion matrix formula that accounts for all classes simultaneously. This ensures the reported MCC remains a balanced and robust measure of model performance, even when more than two classes are involved (Gorodkin, 2004; Chicco and Jurman, 2020). Additionally, balanced accuracy was computed as a secondary metric:

$$\text{Balanced Accuracy} = \frac{\text{Sensitivity} + \text{Specificity}}{2}$$

The optimal model configuration was selected based on the highest cross-validated MCC score, ensuring that the final model provides the best balance between predictive accuracy and model complexity while appropriately accounting for the ordinal nature of the response variable. The analysis was conducted in *R* (R Core Team, 2024) using the `caret` package framework (Kuhn, 2015).





## Appendix C: Matrix usage - warning service groups B and E

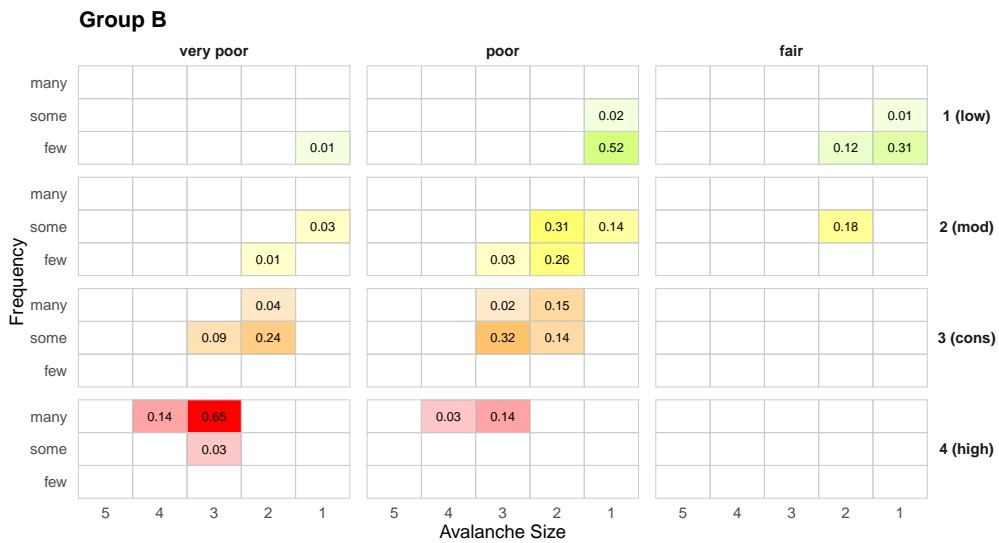

**Figure C1. Dry-snow avalanche problems - warning service group B (Table 1):** Proportion of cases that a specific danger level (rows: $D = 1(\text{low})$ to $D = 4(\text{high})$) was used for a specific matrix combination. Colour shading corresponds to the proportion of cases. Cells with $< 1\%$ usage are not shown.

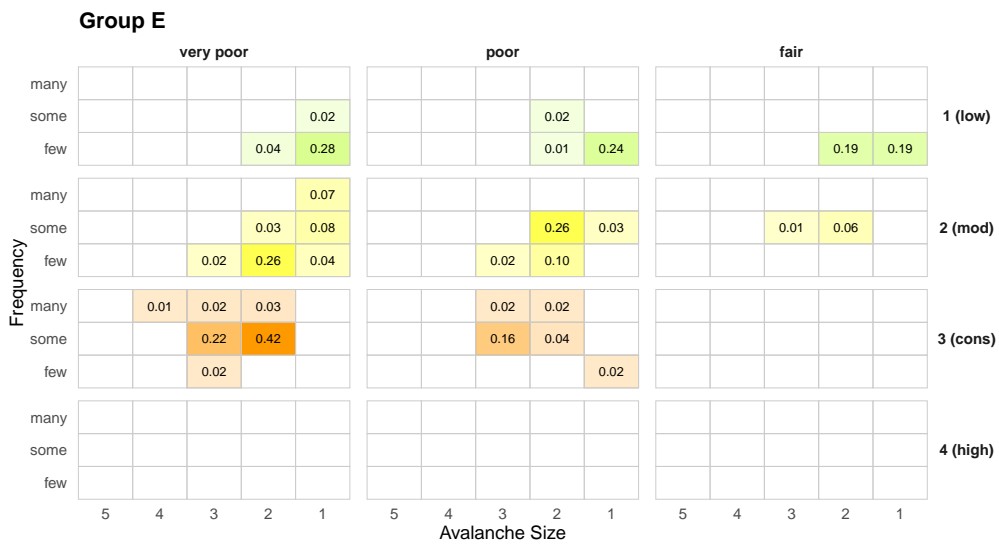

**Figure C2. Wet-snow and glide-snow problems - warning service group E (Table 1)** Refer to Figure C1 for details.




**Appendix D:  Matrix usage regardless of danger level**

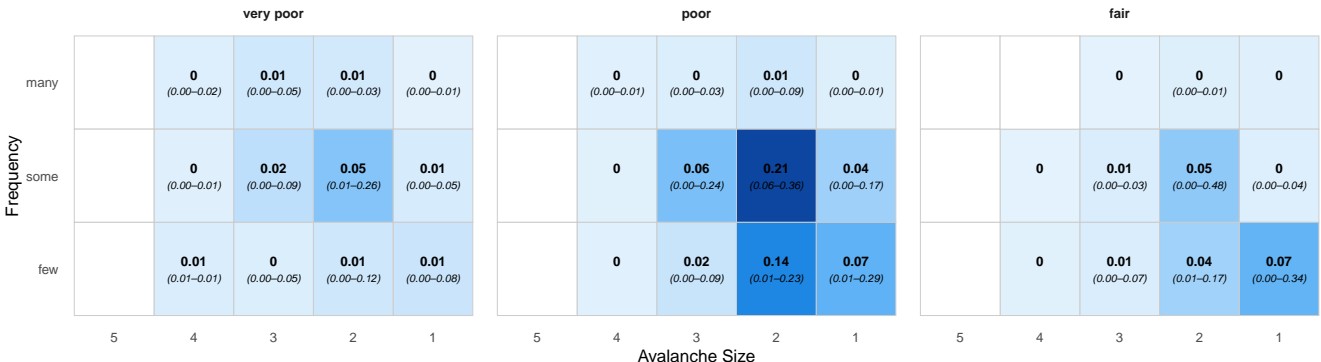

**Figure D1. Dry-snow avalanche problems.** Shown are the median proportion of cell usage of the 26 warning services. The values in brackets represent the min-max range.

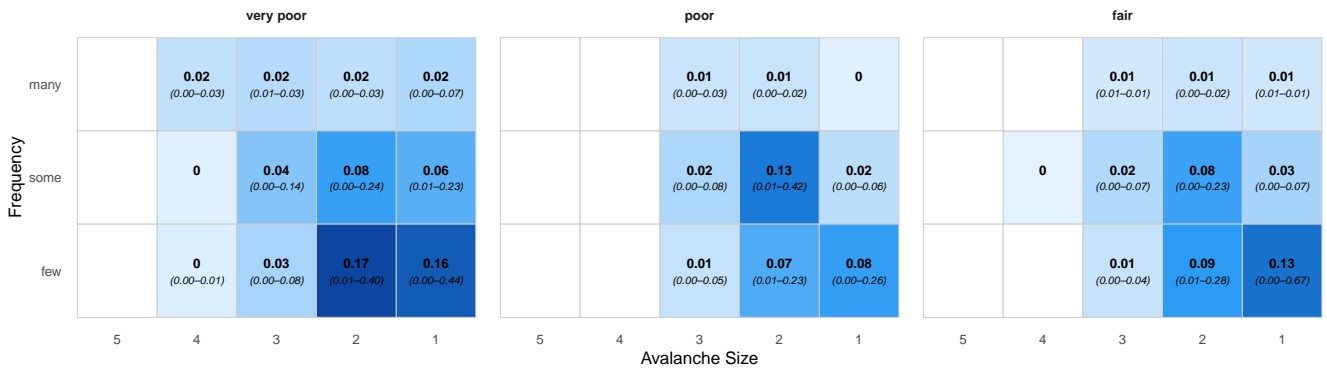

**Figure D2. Wet-snow and glide-snow problems** Shown are the median proportion of cell usage of the 25 warning services (no data for SCO). The values in brackets represent the min-max range.



*Code and data availability.* Data and code will be published at the repository envidat.org, and will also be indexed at https://opendata.swiss/en/.

*Author contributions.* FT (study design, data curation, formal analysis, writing, reviewing), KM (project lead, reviewing), CMa (study design, formal analysis, writing, reviewing), CMi (study design, writing, reviewing).

*Competing interests.* We declare no conflict of interest.

*Acknowledgements.* We thank Stefano Sofia, Petter Palmgren, Nicolas Roux, Giacomo Villa, Guillém Martin Bellido, and Lorenzo Bertranda for invaluable discussions within the EAWS working group *Matrix & Scale*. Filip Kyzek, Mark Diggins, Igor Chiambretti, Giacomo Villa, Guillém Martin Bellido, and Lorenzo Bertranda provided data. We further thank editor Pascal Haegeli for his valuable feedback on the initial manuscript, which ultimately led to the decision to separate the conceptual development (Müller et al., 2025) and the operational analysis

(this study) into two publications. We acknowledge the use of *ChatGPT-4o* (OpenAI) to support language editing of this manuscript and to assist with debugging of the R code.





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
