# Peer review of "The EAWS matrix, a look-up table to determine the regional avalanche danger level (Part B): Operational testing and use"

_EGUsphere, 2025_

## Referee Comment (RC1)

Review of *The EAWS matrix, a look-up table to determine the regional avalanche danger level (Part B): Operational testing and use* by Techel et al. (nhess-2025-3349)

**General Comments**
In this manuscript, the authors present the second part of the EAWS Matrix development process – analysis of operational implementation. The first part (A) in this pair of manuscripts, which I also reviewed, describes the conceptual development of the Matrix. Here, part (B), the authors examined Matrix usage and implementation across avalanche warning services throughout multiple countries. They analyzed the three factor classes of snowpack stability, frequency of the lowest snowpack stability class, and expected avalanche size and, ultimately, the danger level from 26 warning services over three seasons. They presented results aggregated across all groups as well as segregated by avalanche problem categories (dry snow and wet snow/gliding). Finally, they used more detailed data from specific services to assess whether specific actions in any given service helped explain differences in assigned danger level where there were sometimes two assigned danger levels.

I reviewed the original submission prior to portioning the work into two separate but conceptually linked manuscripts. The authors adequately addressed my original comments. I think the new structure of a pair of manuscripts improves the readability and organization. I also think the new analysis provides a more direct assessment of the efficacy of the Matrix across many warning centers and multiple seasons, rather than individual forecaster assessments throughout one season. The manuscript is well organized. The methods are appropriate for the data. The results are clearly presented with informative and effective figures, and the discussion provides sufficient interpretation of these results. I think the limitations are clearly presented in the beginning of the discussion and in the dedication section (6.3). I only have a few minor questions/suggestions for the authors to consider. Overall, I think this is a very worthy contribution to the field of avalanche forecasting and improves our knowledge and use of forecasting tools.

**Specific and Technical Comments**
What is the reasoning for separating dry snow with wet snow problems for the analysis. Clearly, differences exist, but providing some explanation as to why you segregated based on this in section 4.2 would help. For example, you could have also partitioned within dry snow problems to examine differences between new snow/wind slab and persistent slab problems.

Lines 416-419: By recommending three specific cells be reduced to one danger level, do you think that would preclude some forecasters from using the nuance and local knowledge of the warning service that you discuss earlier that leads to using either level 3 or 2 for those cells? In other words, does reducing the choice to 1 danger level exacerbate the problem of forecasters changing their factor assessments (a potential outcome that you mention in lines 355-356) in order to reach the now removed danger level?

Line 2-3: Second sentence in abstract isn't a complete sentence. Consider "To promote greater consistency…, a revised version of the EAWS Matrix– a structured…- **was developed.**"

---

## Referee Comment (RC2)

**Review of Techel et al. (2025): The EAWS matrix, a look-up table for the avalanche danger level (Part B): Operational testing and use**

Benjamin Reuter1

¹Météo-France, Direction des opérations pour la prévision, Coordination Montagne et Nivologie, Saint-Martin d'Hères, France

Correspondence: Benjamin Reuter (benjamin.reuter@meteo.fr)

**1 Summary and recommendation**

The presented article describes an evaluation of a recently developed look-up table for the avalanche danger level during operational use. Results from 26 avalanche forecasting services were compiled and analyzed. The results stimulate ongoing discussion on (the use of) concepts in avalanche forecasting. Finally, suggestions to modify the table are made. There is no doubt that the present work is important for the European avalanche forecasting community – and may in the mid-term improve forecasting consistency and transparency. Some of the article's key points:

- For most of the cases, the authors found good agreement of issued and matrix-suggested danger levels. However, this could be expected as again the "same" population, i.e. avalanche forecasters, were questioned only this time, in a real-life forecasting setting.
- Results in some cells of the table, and in two cells in particular, were more dispersed than in other dry-snow situations. The authors provide hypotheses, but a clear explanation for the observed differences does not seem in reach, yet.
- More importantly, wet- and glide-snow avalanche situations led to considerable differences between issued and
  matrix-suggested danger levels. It seems that the look-up table, and in particular the concepts behind, could not
  be applied with sufficient rigor. Vague stability descriptions and other workarounds may be at the origin of the
  observed differences.
- A finer resolution for some cells is demonstrated based on evaluations from 2 forecasting services. The value of
  finer resolution remains unclear for risk management and applicability in forecasting services. The look-up table
  relies on a classification concept with discreet and clear-cut classes, after all.
- The suggested simplification of the look-up table can be regarded a practical outcome. Some cells were only rarely
  used during the 2 winters of testing. Some cells showed a majority vote for one choice of danger level.
  Modifications are suggested but the final result need to be published.

Some questions arose during the review regarding methodology and interpretation of the results. Improved descriptions of stability and frequency classes seem of major importance for the success of the project. I structured my comments into major comments, which address the analysis and the interpretation of the results, and minor comments related to presentation and writing.

Overall, the study seems suitable for *Natural Hazards and Earth System Sciences*. I hope the comments are helpful in bringing the manuscript to publication.

**2 Major comments**

**2.1 Independent data**

Building on expert opinion is an obvious way forward to create a decision aid. The EAWS matrix or look-up table is being developed this way. While one may argue that this approach may also lead to better acceptance in the forecasting community, cultural differences between forecasters due to empirical local knowledge in the Alpine countries (UNESCO intangible cultural heritage), errors from data interpretation (during temporal or spatial extrapolation from point data or when treating uncertainty) or discrepancy in concept application (e.g. stability classes for dry- or wet-snow situations) introduce uncertainty along the way (of building the desired decision tool). Some of the limitations are mentioned in section 6.3., but readers may wonder how the present study deals with or even mitigates the issues.

The authors state in the Introduction (L49) and in the Discussion (L337) that the assessed parameters are not measurable. This lack of an "independent reference" (L337) is somewhat true for the danger level, but less so for the input parameters of the look-up table. Frequency distributions were derived from avalanche observations, stability tests or snow pits (including tests). A classification for dry-snow stability classes is available in Schweizer et al. (2021). Hence, examples of typical avalanche situations and their respective stability/frequency classes − and possibly, independent data to verify the danger level, are available. Such data will not do the job of choosing the final danger level (which remains the goal of the look-up table), but uncertainty in data interpretation, and difficulty in concept application are mitigated; possibly making for useful complement data. These data should not go unmentioned, and even be considered for improving the descriptions of the frequency classes and stability classes (→ other comment).

The authors explain their choice of methodology, which is appreciable. Nevertheless, the article can benefit from clearer arguments and some adjustments regarding the methodology. Hence, the authors are encouraged to include some data from research publications describing the triplet stability/frequency/size and the (forecast/verified) avalanche danger level.

- Data from avalanche observations (Schweizer et al., 2020), stability tests (Techel et al., 2020), and possibly other
  data such, possibly (stratified) numerous field measurements, may help to corroborate / discard danger level
  choices in some fields. The challenge of choosing a danger level will persist, if the data lack unambiguous
  evidence, but due to interpretation of large data and rigorous concept application the results will be dependable.
  They can likely resolve the most classic situations and fix a (single) danger level.
- Well-documented situations of the past can illustrate this article. To do so, data need to be compiled to choose
  factor classes and independently, fix the danger level. Presenting these situations in the article (very synthetic
  and schematic presentation, supplementary material?) will improve the readers' understanding of the forecasting
  challenges. Furthermore, these situations will serve the forecasting community as benchmarks when familiarizing
  with the concepts or in trainings.

Such situations should cover a wide range of classic situations covering dry-, wet- and glide-snow situations, include skier-triggering only, natural release etc. They should provide the key for a user of the look-up table to what a classic "poor-some-2 = moderate" looks like (in hindsight with the best possible data available).

**2.2 Two levels in one field**

Readers may wonder why the final table does still show up to two levels per field with one of them not corresponding to the color of the field. Wasn't the goal of the desired decision aid to promote consistency in the choice of the danger rating? Don't optional choices invite for deviation from the concept?

Could the authors provide in their manuscript a final version of the look-up table highlighting the suggested changes? (the avalanches.org webpage seems to show a somewhat updated version)

Moreover, why are secondary danger levels provided in the white/no-shaded fields that are considered rare/implausible situations. If data are sparse, then shouldn't those fields simply not show a rating or shouldn't only one rating be suggested in parenthesis to indicate implausibility?

Discrete levels are inherent to classifications. As the authors state, the benefit of the look-up table lies within improving consistency by promoting concepts in the forecasting community. This is to a large part due to the classification's simplicity and not due to the ultimate level of detail the classification allows for. The simplest table with clear-cut classes would come closest to this goal. Here are two supporting arguments:

- The danger level sets the stage in the risk management strategies but alone, will never make up the decision. (Effective risk management depends on understanding the characteristics of an avalanche situation probably communicated by avalanche problems, factor classes, ...). Users do not need a single number "that does it all", but rather a transparent reasoning behind a danger level consistent with the additional information possibly provided (e.g. factor classes).
- The possible resolution of the factor estimates (and finally the danger level) varies across forecasting services and regions. The look-up table need to remain applicable for (most) avalanche forecasting services/scenarios/regions. A small number of well-defined classes seems to be key again.

If sub-classes are introduced or fields further divided (Figure 8) the classification concept is being eroded and the ultimate goal may move out of sight. To this end, I encourage the authors to reflect on the use of the classification concept and the required level of detail. In this context suggestion to refine the matrix made in L434 should be given a second thought and possible disadvantages and constraints (see above) mentioned.

**2.3 Definition of stability and frequency classes used in the study**

The presented results for wet and glide-snow situations show a large spread, in particular with respect to the stability classes (see use of "very poor"). As current definitions, in particular for wet-snow situations (tables in appendix), lack tangible elements, it is no surprise that results are somewhat inconsistent. In Europe, traditions to deal with avalanche hazard vary between countries/cultures. Hence, unambiguous definitions are paramount and will condition any multi-cultural evaluation in Europe.

The reference documents offer plenty of room for improvement regarding factor estimation. In the following, I try to provide some starting points to revise in particular the definitions of the classes shown in the annex documents (and on the avalanches.org webpage?).

Table A1, describing point scale snow stability classes, provides little conclusive information (definition of "difficult," "easy", "very easy"?), is miss-leading ("natural" cannot be a special case of "very easy to trigger") and lacks information to assess stability classes in wet and glide snow situations.

To provide some examples, the word "trigger" (which relates to artificial triggering) appears in the class "very poor", next to an example "natural". According to the table caption, the table refers to the point scale, however, the observations of avalanche release mentioned in the column "description" happen at the slope scale (see use of key words artificial triggering or natural release). What is the relevant stability class in wet-snow situations (no mention in the table)? Point releases only, linear fracture lines expected, "skier sluffs" only, secondary release of persistent weak layers? The authors state in the article that if natural wet-snow activity is forecast, they expect the category "very poor" to be chosen. In a number of cases, however, wet-snow activity can be exclusively limited to point releases. Then, the snowpack stability class can be "good/fair"

as weak layers are not present in the snowpack and humidification leading to avalanches with linear fracture lines (crowns) can be ruled out. How would this situation be treated in terms of snow pack stability? I assume that those avalanche situations could be separated by size, which is, however, not a first-level entry parameter to the matrix. The referenced document (EAWS 2025a) mentions "artificially triggered wet-snow avalanches" in the "(very) poor" category for wet-snow conditions. Either the avalanche type or the release mode is to be changed here.

Along the lines of the definitions of snow instability on the EAWS webpage, table 1 should

- Separate dry, wet and glide snow problems,
- Explicitly refer to avalanche types (including point releases) when providing examples,
- Clearly distinguish natural release and artificial triggering, and most importantly
- Provide tangible snowpack descriptions (see table A1 for dry-snow instability in Schweizer et al., 2021)

A description solely based on tests (and omitting) stratigraphy is difficult in a forecasting context, as stratigraphy is more easily extrapolated and is less variable in time and space, ... and is probably also more widely available in forecasting services.

Table A2, which describes frequency classes, should be revised. The definitions of the classes "some" and "a few" lack tangible descriptors. In the second line describing the class "some", for instance, points in the terrain won't be "many" or "a few", which are the descriptors of the neighboring classes! Moreover, describing a qualitative descriptor, like "a few" with another one, such as the descriptor "rare" does help for application – in particular if they do not refer to the same quantity. Please consider frequency definitions in related fields such as IPCC or also medical sciences (key word rare diseases). Descriptions should include:

- Examples from validation studies where point instability frequencies were documented (Schweizer et al. (2003), Schweizer et al. (2020), Techel et al. (2020)...)
- Examples with dry snow natural release (i.e. stability class very poor) where the frequencies are known. Those are the only situations where "all" points were tested in the terrain. Similar data are available for natural wet-snow release (possibly separation by avalanche types necessary, i.e. point release / linear fracture lines).

Table A3 provides descriptors useful for communication with a wider public, but is too vague for conducting a study with experts in the field of avalanche forecasting. The column should provide descriptions which are useful in a forecasting setting, such as expected avalanche type (point releases / linear fracture lines, skier triggering only, natural release, ...) or typical release volume (usually estimated from forecast stratigraphy). In this way, situations can be better constrained, and choices become easier. An additional column should provide examples such as: Point releases due to snow surface wetting will hardly ever exceed a size2 due to the limited volume in the release area. Stratigraphy with around 50 cm of hardness first in starting zones, can easily exceed avalanche size 2.

**2.4 Compliance with the suggested danger level**

Another strong point of the article is that the authors managed to shed light on how the forecasting services comply with the matrix.

- In summary LL 264 -275 say: For the combination poor-some-size2 group C chose about as often danger level "2-moderate" as danger level "3-considerable. Group A chose danger level "2-moderate" 4 times more often for the same factor combination. This is a discrepancy, but even more alarming are the choices for the combination very poor-somesize3: group C chose 2.5 times more often danger level "4-high" than danger level "3-considerable", but the choices of group A are inversely related: about 4 times more often danger level "3-considerable" than danger level "4-high".

Monitoring matrix compliance seems to be an interesting path for forecasting services to identify diverging situations. If they manage to identify and train they can increase their forecasting quality/consistency. Could be included in conclusions. For now this subjected is touched on in the discussions section in L430.

**2.5 Avalanche size**

Avalanche size has been recently been identified as a relevant element, but as an element of secondary importance in danger level assessments. Avalanche size is particularly important to distinguish between level considerable and high (e.g. Techel et al. 2020, Schweizer et al. 2020). Only rarely it can be decisive between levels "1-low" and "2-moderate" and "3-considerable". In the current version of the look-up table avalanche size seems to play a more important role. For a given stability class ("very poor" or "poor") and frequency class (e.g. "some") the decision on avalanche size determines the choice between levels "4-high", "3-considerable" and "2-moderate" and possibly even "1-low".

How can the structural choice of the look-up table be corrected for reconciliation with non-negligible research results that were obtained from a wide range of conditions and large data.

**2.5.1 Analysis**

- 176: "We then computed the proportion of disagreement between the forecast and Matrix-derived danger levels across different danger levels, avalanche problems, and warning services [...]." Not clear what has been done. A disagreement rate of danger level estimates (between issued value and matrix-suggested value) was calculated. It seems like the danger levels referred to different reference units (across different avalanche problems??) or that averages were computed (across different danger levels??). Please clarify.
- L198: The assumption that Scottish data represents (mostly) dry-snow conditions should be justified, for instance by snow cover model data or snow climate data. Still, if more than 50% (~mostly) of the data are dry-snow conditions, this is a strong simplification.
- L222: "At danger level 1 (low), only 17% of the cases were described using very poor stability, suggesting that natural avalanches were rarely considered." Neither do 17% correspond to "rare" (see e.g. IPCC frequency definitions), nor does the class "very poor" exclude artificial triggering.
- L223-226: "According to the intended logic [...]. Thus the data implies that neither natural avalanche activity nor human-triggered avalanches were considered relevant in more than half of the cases (56%)." The reader wonders what the avalanche problem finally was. The description of the class "fair" reads: "difficult to trigger", hence one would assume that triggering avalanches is possible in "fair". In general, in danger level "1-low" or "2-moderate" (also 17% of the cases in class "fair") avalanches are still triggered by single people, and not exclusively by explosives or groups, but avalanche prone locations and also trigger locations are just less frequent. Please consider rewording.
- L243: "Across all danger levels, most Matrix cells were predominantly used for a single danger level, [...]" Well, in the end one single number need to be chosen indicating the danger level. Please consider rewording and checking the instances with "single danger level" in the manuscript (other instances, e.g. L259, L347, L417).

- L258: "Most Matrix cells were associated with a single danger level, and overlaps were rare". No surprise, as this is the
  definition of Group A: high compliance. You may even say "in almost all cases" instead of "most".
- L259: "fair-a few-size 1", shouldn't it be size 2?
- L263: "indicating greater overlap in the use of the same factor combinations for different danger levels" (Also see instance in L259.) What it is that overlaps? It looks like several factor combinations (more combinations than the matrix suggests) led to the one (issued) danger level. This happened more often than in Group A which obviously corresponds to the definition of compliance....

**2.5.2 Interpretation**

Comparisons of the obtained results with benchmark situations are appreciated. In some instances, I would like the authors to double check their statements and adapt the wording if needed.

- L225: The conclusion: "Thus, the data imply that neither natural release ..." does not seem in line with the stability classes in table A1. In the class "fair" skier triggering still seems possible, but "difficult". Please consider revision.
- L255 and following: these paragraphs talk about compliance. Please consider to shorten and condense the message.
   Also see section Minor comments.
- L268: "neighboring cells" How do we know? The choice was a different size, frequency class or possibly even stability class? Well, what does "neighboring" actually mean here?
- L276: The mentioned shared patterns (e.g. poor-some-size2 == moderate) seem to be pretty classic situations. Could you make the link to frequently observed situations in research articles such as Techel et al. (2020) or Schweizer et al. (2020) or Schweizer et al. 2003? Can you corroborate the matrix-suggested danger level with even more data from field studies? I assume, if you can do so, a decision will emerge for one single danger level in this field.
- Glide snow avalanches and wet-snow avalanches (linear fracture lines) are exclusively natural releases. The authors recall that statement in L304 and discuss differences between forecasting services. The defining document of EAWS however mentions "artificial triggering of wet-snow avalanches" in the category "very poor poor". Apart from communication/training issues in the forecasting services, I assume that incoherence related to the understanding of avalanche types, release processes and stability classes are at the origin of the differences the authors identify.
- L351: In Figure 1 the mentioned cells correspond to levels "moderate"/"considerable" or "considerable"/"high". Do you mean: all forecasting services from one group assigned the same one danger level to the field? Please improve the wording to be clearer.
- L353: "The reasons for this discrepancy remain speculative. Divergent use..." How can the option be ruled out that forecasters had a danger level in mind when they chose from the look-up table and simply chose a field producing the preferred danger level? Good consistency between forecasting services in many other dry-snow situations suggests that forecasters are able to estimate factor classes with a certain reproducibility among each other. The option of reverse engineering, i.e. "choosing a field representing the preferred level" unfortunately seems quite plausible.

- L375: appropriate level of granularity: Obviously, current risk management methods can neither exploit sub-danger level classes (cf. even in the reduction method the slope angle estimates, probably precise around 2-3°, remain the limiting factor) nor exploit stability classes, frequency classes or size classes directly.
- L382: rarely used cells: Have data from all relevant climate zones been considered? (One could even ask if tests have been done over sufficiently long periods that even less frequent situations could occur sufficiently often to be picked up by the data analysis.)
- L391: "High matrix compliance means narrower set of cells was used". Does "narrower set" refer to Figures 3 and 5?
   If so, the lower spread in the data means that the issued and the matrix-suggested danger levels more often agreed.
   Isn't that the definition of compliance? The conclusion that the "matrix helps standardize assessments" (L394), so yield consistent danger level ratings for the same factors, seems to be based on a circular statement.
- L408: "In our analysis, the most notable differences related to the classification of snowpack stability for wet- and glide-snow avalanche problems. Although these avalanche types are generally associated with natural avalanche occurrence". Avalanche problems and avalanche types are two distinct concepts. Please keep them apart. Moreover, the statement on wet and gliding snow could be more straightforward. Wet and glide-snow avalanches are exclusively due to natural release except for coincidence of human presence. Weak layer wetting and snow gliding do not require human intervention.
- L422: "could also be shaded white to indicate higher uncertainty" Do you mean that these cells are not plausible, as they do not (sufficiently often) occur in nature?
- L453: please explain the term "effective" in this context
- L454: "supporting consistent danger level assessment", please provide evidence, for example "as reflected in our analysis of.../ xy% of the forecasters issued ...."
- L458: inconsistencies in wet-snow situations. The concept of the stability classes (very poor poor fair -good) does not seem to accommodate well wet- and glide-snow situations. In fact, it has been developed for dry-snow stability. The analysis that some forecasting services (group F) almost only use "very poor" to describe wet-snow situations is interesting. The rule of "using very poor for wet-snow", however, only seems to work as all danger levels are present on the panel "very poor".
  - The field with level "1-low" on this panel was chosen in 46% of the wet-snow cases to for level 1 judgements. In dry-snow situations this field is hardly used (~5%). Regarding wet-snow avalanches, the field does not seem as relevant as the percentage may suggest. Typical wet-snow avalanches often have volumes >>100m3 as they start within the snowpack, and rarely come alone due to the widespread nature of the wetting process. Also regarding dry-snow situations this field does not seem to be very relevant (except for a handful days in early winter maybe) as the definition of "very poor" stability (e.g.Schweizer Wiesinger, 2001: profile type: 1, 5, 7 and 9, dominant weak layers of surface hoar or facets; bottom frequently weak with one cohesive slab; RB1/2 whole block) does not go well with "maximum expected avalanche size 1" and "a few locations".

Hence, the field with "1-low" on "very poor" stability essentially seems to be justified by wet-snow situations – and is probably used when marginal avalanche activity / point release only is expected – which does not seem in line with "very poor" stability. Selected well-documented situations and improved stability descriptions will hopefully improve the matrix use for wet snow situations.

L456: "Analyzing finer-granularity factor assessments can reveal tendencies within Matrix fields and may offer a
path toward more specific guidance in cells that currently contain two danger levels ". Finer resolution sounds great,

but it is useful? In the near-future, advances in avalanche forecasting need to strive for clear communication of the pieces of information required in the risk management strategies that have established in the various user groups. The danger level is only the first communication vehicle that sets the stage. Fine-tuning the danger level will not necessarily result in improved risk management performance.

**3 Minor comments**

- Please find one term to address the danger level that was chosen by the forecasters or that was the result of the lookup table (issued, forecast, assigned) to improve reading comfort.
- L47: "Ideally, such evaluation would consider both quality ... ". Consider rewording the examples for "quality" and "consistency". The target variables need to be clearly stated. The term "accuracy" maybe better understood in this context than the term "quality" (even though it's the original term Murphy used), which is often associated with a broader meaning.
- L56+L59: Do you prefer "practical" implementation or "operational" implementation?
- L75: Do you prefer to call them "factors" or "components"?
- L173+176: "matrix-derived" or "matrix-suggested", please identify one single term for the danger level derived with the help of the look-up table and apply it throughout the manuscript
- L174: "...application of the Matrix: "The colon does not introduce the following. Consider adding a sentence like: "the disagreement was computed as ..."
- L218: "[...] specific avalanche problems." I guess you mean dry- or wet-snow conditions in this context.
- L235: rewording suggested: "...stability was as often described by "very poor" as by "poor" [...]"
- Figure 3: This (nice!) figure illustrates factor estimates and (final) issued danger level. The word "use" in the Figure caption is too general to be understood ("matrix use", "was used for a specific..."). Consider using terms like "issued danger level" and "matrix suggested danger level", agreement between "issued and suggested/proposed danger level". Please consider highlighting (black contours around the cells?) the "matrix-suggested" danger levels. All axes need to be clearly indicated, please complete: Stability classes, Issued danger level, Frequency classes
- Figures 5: the same comments as for Fig. 3 apply, please consider them to improve Fig 5 equally
- L257: Is there need to talk about avalanche problems (EAWS definition, new snow, wind slabs, ...) in this article? In this instance, wouldn't the term "dry-snow avalanche situations / conditions" do the job in order to refer to any release on dry weak layers, i.e. to exclude the gliding of the entire snowpack and the water-induced loss of stability? If no, please explain what dry-snow avalanche problems include and how they were determined by the forecasters who participated in the study.
- L257: "characterized by higher compliance" or finish the comparison, e.g. "higher than ...." or "highest compliance"
- L261: "In contrast, Group C defined by Pdisagree ≥ 7%, though presumably characterized by high heterogeneity [...]", please reword as the conjunction word does not work out well

- L266: please explain the observations, for example: poor-some-size2 led to an issued level "3-considerable" in 7 % of the cases, and to "2-moderate" in 38% of the cases in group A, while in Group C choices were more balanced between danger levels "3-considerable" (in 26 % of the cases) and to "2-moderate" (in 27 % of the cases).
- L328 and the following paragraphs of the discussion section: It may be convenient to start a discussion paragraph with
  a concise summary of the observed results.
- L344: "most danger levels were most frequently associated", please be specific
- L417/418: consider replacing "strong" with other adjectives to go with the verb "use", such as: often, frequently, a lot, many times
- L421: consider alternatives for "under-supported", such as lacking data support, underrepresented in our data
- L424: suggest rewording for clarity: "These cells lie in the stability class" fair" and concern all fields of avalanche size 4 or 5, and all fields with frequency many. In the stability class "poor" the fields with avalanche size 5 are concerned and can now lose their shading color."

---

## Author Comment (AC1)

**Response to review by Erich Peitzsch**

Frank Techel, and co-authors

We thank the reviewer for the detailed and constructive reviews of both the initial manuscript, and now, of Part B. We sincerely appreciate the time and expertise invested in these reviews, which helped identify points requiring clarification and improvement.

Below, each reviewer comment is reproduced in gray, followed by our response in blue. Planned revisions to the manuscript are indicated in red.

**General Comments**

In this manuscript, the authors present the second part of the EAWS Matrix development process – analysis of operational implementation. The first part (A) in this pair of manuscripts, which I also reviewed, describes the conceptual development of the Matrix. Here, part (B), the authors examined Matrix usage and implementation across avalanche warning services throughout multiple countries. They analyzed the three factor classes of snowpack stability, frequency of the lowest snowpack stability class, and expected avalanche size and, ultimately, the danger level from 26 warning services over three seasons. They presented results aggregated across all groups as well as segregated by avalanche problem categories (dry snow and wet snow/gliding). Finally, they used more detailed data from specific services to assess whether specific actions in any given service helped explain differences in assigned danger level where there were sometimes two assigned danger levels.

I reviewed the original submission prior to portioning the work into two separate but conceptually linked manuscripts. The authors adequately addressed my original comments. I think the new structure of a pair of manuscripts improves the readability and organization. I also think the new analysis provides a more direct assessment of the efficacy of the Matrix across many warning centers and multiple seasons, rather than individual forecaster assessments throughout one season. The manuscript is well organized. The methods are appropriate for the data. The results are clearly presented with informative and effective figures, and the discussion provides sufficient interpretation of these results. I think the limitations are clearly presented in the beginning of the discussion and in the dedication section (6.3). I only have a few minor questions/suggestions for the authors to consider. Overall, I think this is a very worthy contribution to the field of avalanche forecasting and improves our knowledge and use of forecasting tools.

**Specific and Technical Comments**

- **Segregation of dry vs. wet snow problems:** What is the reasoning for separating dry snow with wet snow problems for the analysis. Clearly, differences exist, but providing some explanation as to why you segregated based on this in section

4.2 would help. For example, you could have also partitioned within dry snow problems to examine differences between new snow/wind slab and persistent slab problems.

– **Lines 416–419:** By recommending three specific cells be reduced to one danger level, do you think that would preclude some forecasters from using the nuance and local knowledge of the warning service that you discuss earlier that leads to using either level 3 or 2 for those cells? In other words, does reducing the choice to 1 danger level exacerbate the problem of forecasters changing their factor assessments (a potential outcome that you mention in lines 355–356) in order to reach the now removed danger level? - This is an important point, and frankly: yes, it may well be that this exacerbates the problem of forecasters changing the factor input to be in line with a danger level. - We'll add a remark along that line in the respective paragraph in the discussion section.

– **Line 2–3:** Second sentence in abstract isn't a complete sentence. Consider:
"To promote greater consistency. . . , a revised version of the EAWS Matrix – a structured. . . – was developed." - Thank you for pointing this out. We'll change accordingly.

---

## Author Comment (AC2)

**Response to review by Benjamin Reuter**

Frank Techel, and co-authors

We thank the reviewer for the detailed and constructive review. We sincerely appreciate the time and expertise invested in the review, which helped identify points requiring clarification and improvement.

Below, each reviewer comment is reproduced in gray, followed by our response in blue. Planned revisions to the manuscript are indicated in red.

Where helpful, we refer to our companion paper, Part A (Müller et al., 2025), which provides the conceptual and methodological background for the EAWS Matrix. We would like to note that the published version of Part A has undergone substantial revisions compared to the initial pre-print, which was available to the reviewer at the time of review. As a number of the reviewer's comments touch on aspects addressed there, we point out explicitly where these issues are treated in Part A.

**2 Major Comments**

**2.1 Independent data**

Building on expert opinion is an obvious way forward to create a decision aid. The EAWS matrix or look-up table is being developed this way. While one may argue that this approach may also lead to better acceptance in the forecasting community, cultural differences between forecasters due to empirical local knowledge in the Alpine countries (UNESCO intangible cultural heritage), errors from data interpretation (during temporal or spatial extrapolation from point data or when treating uncertainty) or discrepancy in concept application (e.g. stability classes for dry- or wet-snow situations) introduce uncertainty along the way (of building the desired decision tool). Some of the limitations are mentioned in Section 6.3, but readers may wonder how the present study deals with or even mitigates the issues. - It is not the objective of this study to mitigate the sources of uncertainty described by the reviewer. Rather, our aim is to use available data on Matrix usage from many European warning services to identify potential inconsistencies in how the Matrix is applied. As we emphasize in Section 6.1 (Interpretation) and Section 6.3 (Limitations), the presented approach can highlight inconsistencies but it does not allow us to determine their underlying causes, which may range from cultural differences to interpretation practices or conceptual discrepancies (L373-381, L404-414). Recommendations supported by the data and our findings are summarized in Section 6.2 (L415-437). These fed directly into Part A (Müller et al., 2025), where they were taken up. For instance, limitations of the Matrix approach and possible ways forward are discussed in detail in Part A, Section 6.3.

The authors state in the Introduction (L49) and in the Discussion (L337) that the assessed parameters are not measurable. This lack of an "independent reference" (L337) is somewhat true for the danger level, but less so for the input parameters of the look-up table. Frequency distributions were derived from avalanche observations, stability tests or snow pits (including tests). A classification for dry-snow stability classes is available in Schweizer et al. (2021). Hence, examples of typical avalanche situations and their respective stability/frequency classes – and possibly, independent data to verify the danger level, are available. Such data will not do the job of choosing the final danger level (which remains the goal of the look-up table), but uncertainty in data interpretation, and difficulty in concept application are mitigated; possibly making for useful complement data. These data should not go unmentioned, and even be considered for improving the descriptions of the frequency classes and stability classes. - We thank the reviewer for highlighting the distinction between (i) the measurability of the danger level and (ii) the availability of observational data relevant to the Matrix input parameters. To clarify our position, we structure our response around three points:

– The stability classes presented in Schweizer et al. (2021), originating from Schweizer and Wiesinger (2001), and related work (e.g., Techel et al. (2020b)), are derived from the interpretation of stability tests and snowpack stratigraphy. Such data is valuable for defining frequency classes and was therefore used to define the four stability classes, when developing the Matrix in 2022 (see Figure 2). Based on large data sets of stability tests, Techel et al. (2020a) derived data-driven frequency class thresholds (EAWS, 2025, p.13), which were also provided as examples in the document describing the determination of a regional avalanche danger level using the Matrix (EAWS, 2025). However, in a forecast setting, stability must be anticipated from other data. To link the rather abstract stability classes to typical field observations, Figure 2 provides a range of examples. We believe that forecasters apply the range of examples given there.

– We agree that observational datasets as explored in the studies by Schweizer et al. (2021, 2020b); Techel et al. (2020a, 2022)) are valuable. However, even if such data were abundant and available for every day and every warning region - which they are not - assigning frequency classes such as *a few*, *some*, or *many* would still require an element of expert judgment. For this reason, the absence of a fully independent reference for the Matrix inputs persists, especially in a forecast setting where no field data may be available on the day of issuing the forecast.

– We agree that data-driven approaches can help refine class descriptions. Looking ahead, data-driven approaches may become increasingly useful. In particular, stability and frequency distributions derived from distributed snow-cover simulations - as demonstrated in Herla et al. (2024b, a) - offer promising avenues for providing spatially consistent, independent estimates of Matrix input classes.

Please note that existing observational and test-based datasets have already been integrated to contextualize the stability classes. They may also be used to further refine frequency class definitions. - We will add a remark regarding the potential use of distributed snow-cover simulations for providing the necessary real-time data for stability and frequency estimation in a forecast setting.

The authors explain their choice of methodology, which is appreciable. Nevertheless, the article can benefit from clearer arguments and some adjustments regarding the methodology. Hence, the authors are encouraged to include some data from research publications describing the triplet stability/frequency/size and the (forecast/verified) avalanche danger level.

– Data from avalanche observations (Schweizer et al., 2020b), stability tests (Techel et al., 2020a), and possibly other data such as (stratified) numerous field measurements, may help to corroborate / discard danger level choices in some fields.

– Well-documented situations of the past can illustrate this article. Presenting these situations (synthetically or in supplementary material) will improve the readers' understanding of forecasting challenges.

Such situations should cover a wide range of classic cases (dry, wet, glide snow; skier-triggered and natural). They should provide the key for a user of the look-up table to what a classic "poor–some–size 2 = moderate" looks like.

We agree that well-documented situations and a comparison with additional data sources can be valuable for illustrating Matrix usage and related danger-level choices. However, incorporating such case material in detail is beyond the scope of this article, which focuses on detecting deviations and inconsistencies in Matrix usage across European warning services based on the available data. We like to mention that an example of the kind requested is provided in Part A, Section 6.2 in Müller et al. (2025), where an example of applying the Matrix is shown. Part A (Appendix A: Figures A1–A3) links typical field observations to dry-snow, wet-snow, and glide-snow stability classes. These examples serve the illustrative purpose suggested by the reviewer and offer users concrete reference cases spanning different avalanche types.

**2.2 Two levels in one field**

Readers may wonder why the final table still shows up to two levels per field with one of them not corresponding to the color of the field. Wasn't the goal of the desired decision aid to promote consistency in the choice of the danger rating? Don't optional choices invite for deviation from the concept? Could the authors provide a final version of the look-up table highlighting the suggested changes? (the avalanches.org webpage seems to show an updated version).

When undertaking this study, it was hoped that clarity would emerge regarding the cells with two danger ratings. However, as discussed in Section 6.1 (L335–343), there are limits to how much the Matrix can be refined now that it is in operational use. Based on the data, we provide several recommendations (L415–424), which fed directly into Part A, where the Matrix is revised. However, for many cells, we had no evidence to propose changes to the indicated danger levels. The affected cells and the resulting updated version are shown there (Figure 3b, Part A) and on https://www.avalanches.org/standards/eaws-matrix/. The continuing presence of up to two levels in some cells reflects uncertainty in the survey outcome and this study's findings, rather than an intention to promote optionality.

Moreover, why are secondary danger levels provided in white/no-shaded fields that are considered rare/implausible situations? If data are sparse, shouldn't those fields simply not show a rating or only one rating in parentheses?

As outlined in Section 5, Part A, the danger levels shown in the white cells had limited support in the survey. Due to a lack of data, these cells could also not be validated using the data in this study. We therefore cannot confirm or reject the indicative

ratings obtained when developing the initial version of the Matrix (survey described in Section 4 in Part A). Instead, and this is the approach taken in Part A, additional cells are coloured white precisely to highlight the uncertainty attached to the proposed danger level(s) in those Matrix cells (see Figures 3a and 3b, Part A).

Discrete levels are inherent to classifications. As the authors state, the benefit of the look-up table lies in improving consistency by promoting concepts in the forecasting community. This is to a large part due to the classification's simplicity and not due to the ultimate level of detail the classification allows for. Two supporting arguments:

1. The danger level sets the stage for risk-management strategies but alone will never make up the decision. Users need transparent reasoning rather than a single number "that does it all".

2. The possible resolution of factor estimates varies across forecasting services and regions. A small number of well-defined classes seems to be key.

If sub-classes are introduced or fields further divided (Fig. 8), the classification concept is being eroded and the ultimate goal may move out of sight. The authors are encouraged to reflect on the required level of detail. Suggestions to refine the matrix (L434) should be reconsidered with possible disadvantages mentioned. - There are two aspects to this: simplicity is one, which calls for few, well-defined classes. This is supported by the fact that humans can generally estimate only a small number of categories reliably (e.g., Miller, 1956; Kahneman et al., 2021), and it aligns with the conceptual clarity and reproducibility of the Matrix approach. However, forcing forecasters to commit early to a single discrete class is analogous to rounding intermediate results, which can discard relevant information and create discontinuities in the outcome.[1] At the same time, humans are comparatively good at assigning relative rankings, i.e. using sub-classes within absolute classes (e.g., Kahneman et al., 2021). Therefore, combining absolute and relative judgments (as shown in Figure 2b) is a promising path that should be explored. - We'll expand L373-382 with the explanation given before.

To further elaborate on this: Data from the 2023/2024 forecasting season (Techel et al., 2024) showed that forecasters in Switzerland agreed on the exact same factor class only 60–65% of the time, but in more than 90% of cases the discrepancy was less than one full class. More importantly, they were undecided between two adjacent classes 20–25% of the time. Incorporating this structured uncertainty by considering two neighbouring cells in the Matrix when between two classes, avoids premature "rounding" (or forcing to select one class) while keeping the final classification simple. This approach, described in Section 6.1, Part A, preserves the benefits of the EAWS Matrix while keeping unavoidable uncertainty explicit until the final step, where choosing a single danger level is necessary for communication. Whether splitting Matrix cells, as we show in Figure 8, provides a way forward remains open for discussion. For instance, when revising the Matrix in Part A, this approach was not taken up.

**2.3 Definition of stability and frequency classes used in the study**
* * *
[1] e.g., Guidance for laboratory analysis: «Do not round intermediate calculations; rounding intermediate values can cause rounding errors in the final results and should only take place after the final expanded uncertainty has been determined» https://nvlpubs.nist.gov/nistpubs/ir/2019/NIST.IR.6969-2019.pdf

**Table 2.** Snowpack stability classes referring to the point scale and the type of triggering typically associated with these classes. For further details, including typical observations related to each class, see Sect. A1 or EAWS (2025). Values in parentheses indicate that the trigger or evidence is not typical but may occur.

| Snowpack stability | Description | Sensitivity (CMAH) | Natural avalanches | Human triggers | Explosive/ Cornice fall | Other indicators of instability |
|---|---|---|---|---|---|---|
| Very poor | Very easy to trigger | Touchy | yes | yes | yes | Shooting cracks, whumpf sounds |
| Poor | Easy to trigger | Reactive | no | yes | yes | (Shooting cracks, whumpf sounds) |
| Fair | Difficult to trigger | Stubborn | no | (yes) | yes | |
| Good | Stable conditions | Unreactive | no | no | no | |

**Figure 1.** Observations related to stability classes used in the EAWS Matrix. Screenshot of table from (Müller et al., 2025).

The presented results for wet and glide-snow situations show a large spread, in particular with respect to stability classes (see use of "very poor"). As current definitions, especially for wet-snow situations, lack tangible elements, it is no surprise that results are somewhat inconsistent. In Europe, traditions to deal with avalanche hazard vary between countries/cultures. Hence, unambiguous definitions are paramount and will condition any multi-cultural evaluation in Europe. Table A1, describing point-scale snow-stability classes, provides little conclusive information (definition of "difficult", "easy", "very easy"?), is misleading ("natural" cannot be a special case of "very easy to trigger") and lacks information to assess stability classes in wet and glide snow situations. - Additional practical observations and a direct link to the terminology used in the CMAH (Statham et al., 2018) have been added to the corresponding table in Part A. As illustrated in Figure 2 (screenshot taken from revised Table in Part A), the revised version clarifies how *very poor* stability relates to natural release without excluding human triggering. - We will replace the table in the Appendix with the updated version from Müller et al. (2025).

Along the lines of the definitions of snow instability on the EAWS webpage, Table 1 should:

– Separate dry, wet and glide-snow problems,

– Explicitly refer to avalanche types (including point releases),

– Clearly distinguish natural release and artificial triggering,

– Provide tangible snowpack descriptions (see Schweizer et al., 2021).

Regarding the request to add specific examples to dry-snow, wet-snow, and glide-snow conditions, such examples were provided for dry-, wet-, and glide-snow avalanches together with the introduction of the Matrix in 2022. These concrete examples complement the stability description. These examples are also shown in Appendix A (Part A, Figures A1–A3; see screenshot of these figures in Figure 2). In addition, the revised Table 1, Part A already provides more practitioner-oriented examples for each of the stability classes.

We acknowledge that the descriptors for the frequency classes are broad and therefore a potential source of uncertainty, as also recognized by the EAWS working group who originally defined them. While this clearly warrants further discussion and future refinement, ideally supported by more data-driven approaches, we do not revise these definitions within this paper. Table A3 presents the officially accepted EAWS avalanche size classification. These definitions are well established opera-

[Figure]

**Figure A1.** Dry snow conditions: Common evidence or indications for snowpack stability classes focusing on dry-snow slab avalanches. Arrows indicate that existence towards lower stability classes is imperative. Natural avalanches are a clear indication for the class very poor, while a low and a high additional load are considered approximately equivalent to poor and fair stability. Observations and stability test results should be regarded as indicative only. Abbreviations: Extended Column Test (ECT), Rutschblock (RB), whole block (wB), partial release (pR). Schweizer et al. (2020a), Techel et al. (2020), +single skier not falling, ski-cut, ++single skier falling, group of skiers, person on foot.

[Figure]

**Figure A2.** Wet snow conditions: Common evidence or indications related to wet-snow stability. If no liquid water is present in the snowpack, wet-snow avalanches are not possible.

**Figure A3.** Glide snow conditions: Common evidence or indications related to glide-snow stability. Glide-snow avalanches are not possible if there is no liquid water present at the snow-spoil interface.

**Figure 2.** Observations related to stability classes used in the EAWS Matrix. Screenshot of figure shown in (Müller et al., 2025).

tionally, and we therefore retain them unchanged here. For expert use, EAWS already provides additional, forecasting-relevant descriptions on https://www.avalanches.org/standards/avalanche-size/.

**2.4 Compliance with the suggested danger level**

A strong point is that the authors shed light on how forecasting services comply with the matrix. Monitoring matrix compliance seems an interesting path for forecasting services to identify diverging situations. If they manage to identify and train, they can increase forecasting quality and consistency. This could be included in the conclusions (currently touched on at L430). - We'll briefly take this up in the conclusions.

**2.5 Avalanche size and methodological clarity**

Avalanche size has been identified as a relevant but secondary element in danger-level assessment. In the current matrix it seems to play a more important role. - We are aware of this research (Schweizer et al., 2020b; Techel et al., 2020a). Comparing the data-driven matrix shown in Techel et al. (2020a) with the EAWS Matrix shown in a slightly different layout (Figure 6b in Part A) shows many similarities.

**Analysis.**

Several specific clarifications are requested:

- L176: clarify computation of disagreement between forecast and matrix-derived levels. - We'll expand this description to clarify the individual calculations.

- L198: justify assumption that Scottish data represent mostly dry-snow conditions. - As we have no data on this linked to Matrix use, we'll refer to the proportions of dry-snow and wet- or glide-snow problems provided by the director of the Scottish Avalanche Information Service. In the six forecast regions, and excluding the cornice problem, which is frequently given in Scotland, dry snow problems were used between 67% and 90% of the time during the 2024/2025 season (data provided by Mark Diggins, head of the Scottish Avalanche Information Service).

- L222–226: reconsider interpretation of "very poor" stability and "natural activity rare". - In addition to natural avalanches, we will mention "very easy to trigger".

- L243–259: re-word statements where "single danger level" is used; check "size 1 vs size 2". - We'll check whether rewording these two sentences can improve clarity.

**Interpretation.**

Comparisons with benchmark situations are appreciated, but some statements should be double-checked (e.g. L225, L255, L268). Clarify meaning of "neighboring cells"; relate observed patterns to field studies (Techel et al., 2020a; Schweizer et al., 2020a). - The Results section reports solely on factor estimates in a *forecast* setting. We don't analyse field observations in this study; therefore, we do not refer to such studies, except when linking very poor stability and typical release mechanisms for wet-snow and glide-snow avalanches. (L304-306, L409-412). - We will refer to the revised Table providing the stability definitions and examples (see screenshot of this revised Table in Figure 1). We will either clarify the meaning of "neighbouring cells" or replace this with: «cells that differ in one of the factor inputs».

Further notes:

– Glide- and wet-snow avalanches are natural releases; clarify wording. - We will review the manuscript to ensure that glide- and wet-snow avalanches are natural releases, in most cases.

– L351–353: improve clarity of described discrepancies. - We will describe the discrepancies more clearly by stating the specific proportions and referring to the specific figure where these discrepancies can be observed.

– L375–394: avoid circular statements when defining compliance; check interpretation. - We will review and streamline these lines to reduce repetition.

– L408–458: ensure consistent terminology for avalanche types vs problems; reconsider statements on "effective" and "supporting consistent danger-level assessment". - We'll check for consistent terminology throughout.

– L456–458: reflect whether finer granularity actually benefits risk management. - As outlined earlier, there are arguments both for and against using finer granularity in factor assessments. Whether such refinement ultimately improves consistency in danger-level assessment remains an open question. We also note that the Matrix is designed to support *danger* assessment, not *risk* analysis.

**3 Minor Comments**

– Use one consistent term for the danger level chosen by forecasters ("issued", "forecast", or "assigned"). - We'll change throughout to ...

– L47: clarify meaning of "quality" vs "consistency"; perhaps use "accuracy". - Quality is the term used by Murphy (1993), who we cite for this concept. We'll consider adjusting to accuracy, which is what we mean and describe by quality.

– L56 + L59: choose between "practical" or "operational" implementation. - We'll change to "operational" implementation.

- L75: choose between "factors" or "components". - We'll change to input "factors" throughout the manuscript

- L173 + L176: choose either "matrix-derived" or "matrix-suggested". - We'll change to "matrix-suggested" throughout the manuscript

- L174: clarify colon usage ("the disagreement was computed as . . . "). - We'll split into two sentences.

- L218: specify "dry- or wet-snow conditions". - For clarity, we'll repeat the definition from Table 2 and L164-165.

- L235: reword to ". . . stability was as often described by 'very poor' as by 'poor' . . . ". - We believe that our wording is correct. But we leave this decision to copy-editor.

- Figure 3 and 5: improve captions and axis labels; highlight matrix-suggested danger levels. - We'll highlight the Matrix-suggested danger levels. We'll increase font size of the figure axis.

- L257: clarify whether "avalanche problems" or "dry-snow situations" are meant. - As outlined before dry-snow avalanche problems refers to the group of avalanche problems relating to dry-snow conditions (L164-165). We'll check throughout manuscript that we use same terminology as described there.

- L261 + L266 + L328 ff: improve transitions and clarity in discussion. - We'll review phrasing of the transition between these paragraphs.

- L417–424: adjust adjectives and phrasing for clarity ("often", "frequently", "under-supported", etc.). - We'll review whether other adjectives may be more suitable.

**References**

EAWS: Determination of the avalanche danger level in regional avalanche forecasting, https://www.avalanches.org/wp-content/uploads/2022/12/EAWS_matrix_definitions_EN.pdf, EAWS working group Matrix and Scale (working group members: Müller, K.; Bellido, G.; Bertrando, L.; Dufour, A.; Feistl, T.; Mitterer, C.; Palmgren, P.; Roux, N.; Sofia, S.; Techel, F.). last access: 10 Jul 2025, 2025.

Herla, F., Binder, M., Lanzanasto, N., Perfler, M., Widforss, A., Reisecker, M., Müller, K., and Mitterer, C.: Synthesizing regional snowpack stability and avalanche problems in the operational AWSOME framework, in: Proceedings, International Snow Science Workshop Tromsø, Norway, pp. 272–277, 2024a.

Herla, F., Haegeli, P., Horton, S., and Mair, P.: A large-scale validation of snowpack simulations in support of avalanche forecasting focusing on critical layers, Nat. Hazards Earth Syst. Sci., 24, 2727—2756, https://doi.org/10.5194/nhess-24-2727-2024, 2024b.

Kahneman, D., Sibony, O., and Sunstein, C.: Noise: A flaw in human judgment, William Collins, London, U.K., 2021.

Miller, G.: The magical number seven, plus or minus two: Some limits on our capacity for processing information, Psychological Review, 63, 81–97, https://doi.org/10.1037/h0043158, 1956.

Müller, K., Techel, F., and Mitterer, C.: The EAWS matrix, a decision support tool to determine the regional avalanche danger level (Part A): conceptual development, Natural Hazards and Earth System Sciences, 25, 4503–4525, https://doi.org/10.5194/nhess-25-4503-2025, 2025.

Murphy, A. H.: What is a good forecast? An essay on the nature of goodness in weather forecasting, Weather and Forecasting, 8, 281–293, https://doi.org/10.1175/1520-0434(1993)008<0281:WIAGFA>2.0.CO;2, 1993.

Schweizer, J. and Wiesinger, T.: Snow profile interpretation for stability evaluation, Cold Reg. Sci. Technol., 33, 179–188, https://doi.org/10.1016/S0165-232X(01)00036-2, 2001.

Schweizer, J., Mitterer, C., Reuter, B., and Techel, F.: Avalanche danger level characteristics from field observations of snow instability, The Cryosphere, 15, 3293–3315, https://doi.org/10.5194/tc-15-3293-2021, 2020a.

Schweizer, J., Mitterer, C., Techel, F., Stoffel, A., and Reuter, B.: On the relation between avalanche occurrence and avalanche danger level, The Cryosphere, https://doi.org/10.5194/tc-2019-218, 2020b.

Schweizer, J., Mitterer, C., Reuter, B., and Techel, F.: Avalanche danger level characteristics from field observations of snow instability, The Cryosphere, 15, 3293–3315, https://doi.org/10.5194/tc-15-3293-2021, 2021.

Statham, G., Haegeli, P., Greene, E., Birkeland, K., Israelson, C., Tremper, B., Stethem, C., McMahon, B., White, B., and Kelly, J.: A conceptual model of avalanche hazard, Natural Hazards, 90, 663 – 691, https://doi.org/10.1007/s11069-017-3070-5, 2018.

Techel, F., Müller, K., and Schweizer, J.: On the importance of snowpack stability, the frequency distribution of snowpack stability and avalanche size in assessing the avalanche danger level, The Cryosphere, 14, 3503 – 3521, https://doi.org/10.5194/tc-2020-42, 2020a.

Techel, F., Winkler, K., Walcher, M., van Herwijnen, A., and Schweizer, J.: On snow stability interpretation of extended column test results, Natural Hazards Earth System Sciences, 20, 1941–1953, https://doi.org/10.5194/nhess-2020-50, 2020b.

Techel, F., Mayer, S., Pérez-Guillén, C., Schmudlach, G., and Winkler, K.: On the correlation between a sub-level qualifier refining the danger level with observations and models relating to the contributing factors of avalanche danger, Natural Hazards and Earth System Sciences, 22, 1911–1930, https://doi.org/10.5194/nhess-22-1911-2022, 2022.

Techel, F., Lucas, C., Pielmeier, C., Müller, K., and Morreau, M.: Unreliability in expert estimates of factors determining avalanche danger and impact on danger level estimation with the Matrix, in: Proceedings International Snow Science Workshop, Tromsø, Norway, 23-29 Sep 2024, pp. 264 – 271, https://arc.lib.montana.edu/snow-science/item.php?id=3144, 2024.